# A YAP-centered mechanotransduction loop drives collective breast cancer cell invasion

Antoine A. Khalil [1] ✉, Daan Smits [1,7], Peter D. Haughton [2,7], Thijs Koorman [2,7], Karin A. Jansen[2], Mathijs P. Verhagen [3], Mirjam van der Net[1], Kitty van Zwieten[1], Lotte Enserink[2], Lisa Jansen[1], Abdelrahman G. El-Gammal[1], Daan Visser [2], Milena Pasolli[1], Max Tak[1], Denise Westland[1], Paul J. van Diest [2], Cathy B. Moelans[2], M. Guy Roukens [1,4], Sandra Tavares[5], Anne-Marie Fortier [6], Morag Park [6], Riccardo Fodde [3], Martijn Gloerich [1], Fried. J. T. Zwartkruis [1], Patrick WB. Derksen [2] ✉ & Johan de Rooij [1] ✉

Dense and aligned Collagen I fibers are associated with collective cancer invasion led by protrusive tumor cells, leader cells. In some breast tumors, a population of cancer cells (basal-like cells) maintain several epithelial characteristics and express the myoepithelial/basal cell marker Keratin 14 (K14). Emergence of leader cells and K14 expression are regarded as interconnected events triggered by Collagen I, however the underlying mechanisms remain unknown. Using breast carcinoma organoids, we show that Collagen I drives a force-dependent loop, specifically in basal-like cancer cells. The feed-forward loop is centered around the mechanotransducer Yap and independent of K14 expression. Yap promotes a transcriptional program that enhances Collagen I alignment and tension, which further activates Yap. Active Yap is detected in invading breast cancer cells in patients and required for collective invasion in 3D Collagen I and in the mammary fat pad of mice. Our work uncovers an essential function for Yap in leader cell selection during collective cancer invasion.

To start invasion, epithelial tumor cells undergo a transformation into protrusive and motile cells. Multiple transcriptional programs have been suggested to induce breast cancer cell invasion, including epithelial-to-mesenchymal (EMT) and luminal-to-basal transitions[1–3]. The exact mechanisms by which such programs induce invasion are not fully understood. Interestingly, cancer cells with mesenchymal or basal cell characteristics are not invasive in all circumstances. It has become clear that cell extrinsic stimuli play a key role and invasion is strongly affected by the biochemical and physical characteristics of the surrounding extracellular matrix (ECM)[4].

Breast cancers invade as individual cells or multicellular groups, the latter termed collective invasion. Invasive ductal carcinoma (IDC) is a breast cancer type in which tumor cells invade predominantly by collective invasion[1,5,6]. Collective invasion is driven by a small subset of cancer cells that acquire leader cell characteristics[7]. Leader cells emerge from a subset of cancer cells that are localized at the tumor-ECM interface and that extend cytoskeletal protrusions towards the

[1]Center for Molecular Medicine (CMM), University Medical Center Utrecht, Utrecht, The Netherlands. [2]Department of Pathology, University Medical Center Utrecht, Utrecht, The Netherlands. [3]Department of Pathology, Erasmus Medical Center, Rotterdam, The Netherlands. [4]Regenerative Medicine Center Utrecht, University Medical Center Utrecht, Utrecht, The Netherlands. [5]i3S—Instituto de Investigação e Inovação em Saúde, Universidade do Porto, Porto, Portugal. [6]Goodman Cancer Institute McGill University, Depts Biochemistry and Oncology, McGill University, Goodman Cancer Institute, Montréal, Canada. [7]These authors contributed equally: Daan Smits, Peter D. Haughton, Thijs Koorman. ✉e-mail: a.khalil@umcutrecht.nl; p.w.b.derksen@umcutrecht.nl; jderooij@y2y.eu

ECM while maintaining cell-cell interactions with the neighboring tumor cells. In many breast cancer models, luminal cells that express basal cell markers (basal-like cells) including cytokeratin 14 (K14) become the leaders of invasion. Despite the presence of many transcriptionally identical basal-like cells at the tumor-ECM interface, only few become leader cells[1]. Apparently, becoming a leader cell does not only depend on active basal cell transcription, but also on cell extrinsic factors. Collagen I is a major inducer of protrusive activity of breast cancer cells in vitro[1,4] and its density and mechanical stiffness strongly associate with collective invasion in breast cancer patients[8,9]. The mechanisms by which cancer cells mechanotransduce the biophysical cues of the ECM in order to become invasive remain not fully understood. Yap is a transcriptional coactivator that is activated by elevations in ECM mechanical stiffness[10]. In breast cancer, whether Yap acts as an oncogene or tumor suppressor remains unclear and appears to depend on the breast cancer subtype. In triple negative breast cancers, several studies report Yap as an oncogene and its expression/activity enhances tumor progression[11]. Among the established oncogenic effects of Yap is the promotion of cell proliferation and stem cell properties[12–15]. Yap is also involved in the migration of breast and other cancer cells. This was mainly studied in vitro using single cell migration assays[16]. A role for Yap in collective invasion, or the induction of leader cell function has not yet been described.

Here, we used breast cancer models to investigate the relation between Collagen I, invasion, and transcriptional reprogramming at the single cell level by transcriptomics and cell imaging. Our work uncovers a YAP-centered feed forward mechanism that explains the emergence of leader cells from basal-like cells located at the tumor-ECM interface. This process starts by contact with Collagen I, which leads to enhanced YAP activity in basal-like cells. Yap initiates a transcriptional program that includes ECM remodelers to increase Collagen I fiber alignment. Hotspots in Collagen I alignment drive further YAP activation resulting in the emergence of the leader cells that drive collective invasion.

## Results

### Basal-like cells guide collective invasion in IDC
To recapitulate basal-cell guided collective invasion in vitro and in vivo, we used tumor organoids derived from mammary carcinomas that developed in MMTV-PyMT mice[17], a model that mimics key aspects of human IDC histopathology and progression[18,19]. MMTV-PyMT tumor organoids were either embedded in Collagen I or injected into the inguinal mammary fat pad of mice (Fig. 1a). Confocal imaging of K14 (basal/myoepithelial cell marker) and K8 (luminal cell marker) showed that the K14-positive cells are positioned almost exclusively at the tumor-ECM interface (Fig. 1b, c). This includes the outer layer of the tumor (rim) and the tips of the invasive strands in both 3D Collagen I cultures (Fig. 1b) and xenografted tumors (Fig. 1c). K14-positive basal-like tumor cells expressed K8, but at lower levels relative to the K14-negative cells. Compared to the myoepithelial cells lining the mammary ducts, the basal-like cells at the invasive tumor fronts expressed similar levels of K14 but at least 3-fold higher levels of K8 (Fig. 1c, d).

To confirm the presence of basal-like cells during invasion in breast cancer[1], we analyzed the expression of K14 and K8/18 in serial tissue sections from 20 IDC tumors (grade 3) by immunohistochemistry (Fig. 1e–h). We focused on the collective invasion patterns, identified as groups of tumor cells (positive for K8/18) that invade the surrounding stroma (Fig. 1f–h, zoom ins). Most IDC samples analyzed (19/20) contained collective invasion patterns. The invasive patterns in 7 out of 19 samples contained K14-positive cells (Fig. 1g, h) that were located either i) throughout the invasive strands (3 out of 7 of K14-positive samples, Fig. 1g) or ii) almost-exclusively at the ECM interface (4 out of 7 of K14-positive samples, Fig. 1h). The K14-positive cells maintained K8/18 expression (Fig. 1g, h, insets) where some that are located at the ECM-interface show a weaker K8 immunoreactivity

(Fig. 1h, inset and black arrowheads). In short, the MMTV-PyMT organoid model recapitulates basal-like cell led collective invasion in a subset of IDC patients.

### Collective invasion in Collagen I is driven by a subset of basal-like cells at the tumor-ECM interface
We next used the MMTV-PyMT tumor organoids to understand how collective invasion is initiated and decipher the underlying role of the ECM in regulating invasion. First, we characterized the luminal and basal features of the tumor organoids formed from single cells under non-invasive conditions (Basement membrane extract, BME) by analyzing K8 and K14 expression. Confocal immunofluorescence microscopy identified two subtypes of tumor organoids (Fig. 2a): A luminal population (~70% of all organoids) that consisted only of K8$^{HIGH}$K14$^{NEG}$ cells (Fig. 2a (i)) and a mixed population (~30%), in which a central luminal K8$^{HIGH}$K14$^{NEG}$ population was surrounded by a layer of basal-like K8$^{LOW}$K14$^{HIGH}$ cells that formed the organoid-ECM interface (Fig. 2a (ii), arrowheads). The presence of K8$^{LOW}$K14$^{HIGH}$ cells in tumor organoids originating from single cells (Fig. 2a, arrowheads), indicate that a subset of the MMTV-PyMT tumor cells may give rise to organoids containing both luminal and basal-like cells. To confirm that basal-like and luminal cells in MMTV-PyMT organoids may originate from the same cell, we established monoclonal organoids. We observed that single cells from MMTV-PyMT cultures could generate either luminal (K8$^{HIGH}$K14$^{NEG}$) or mixed subtype organoids (Fig. 2b), in line with previous reports using a 4T1 breast cancer cell line[20]. The detection of basal-like cells during the monoclonal MMTV-PyMT organoid formation and under non-invasive conditions in BME is consistent with recent studies[21], and indicates that the presence of basal-like luminal cells is not dependent on pro-invasive stimuli from the ECM (for example from Collagen I).

To investigate how Collagen I induces invasive abilities, MMTV-PyMT organoids grown in BME, were transferred to either BME or Collagen I and cultured for up to 5 days (Supplementary Fig. S1a). In both BME and Collagen I gels, the luminal organoids lacked collective invasion (Fig. 2b-d) and the K14-negative cells at the organoid-ECM interface did not show any detectable protrusive behavior (Fig. 2c, d, magenta arrowheads). Cells at the ECM-interface retained K14-deficiency despite contact with Collagen I fibers for several days. Despite the presence of K14$^{HIGH}$ cells, the mixed organoids were not invasive in BME (Fig. 2c). In line with previous reports[4], the K14$^{HIGH}$ cells were predominantly positioned at the organoid-BME interface, yet they lacked detectable protrusions (Fig. 2b, inset). In Collagen I however, a subset of the pre-existing K14$^{HIGH}$ cells extended cytoskeletal protrusions and formed the leader cells of multicellular invasive strands. This was identified by immunofluorescence of MMTV-PyMT organoids at fixed time points (Fig. 2c, d, arrowheads) and by time-lapse confocal imaging of tumor organoids expressing endogenously tagged K14 (Supplementary Movie 1). The formation of the invasive strands was accompanied by extensive ECM remodeling and pulling forces, as visible by the Collagen I bundling and alignment detected by reflection microscopy (Fig. 2c, d and Supplementary Movies 1 and 2). These results indicate that the presence of basal-like cells and a supportive ECM environment are both essential for cells to emerge as leaders that drive collective invasion.

To further assess the basal and luminal cell identity in non-invasive and invasive conditions, MMTV-PyMT tumor organoids were embedded in either BME (non-invasive) or Collagen I (invasive) for 5 days and analyzed by single cell mRNA sequencing. Dimension reduction with UMAP unsupervised clustering analysis revealed 3 cellular groups. Based on K14 mRNA expression, cluster 1 was defined as a basal-like cluster (BME 26.9±3.5%, Collagen 31.1±0.8% of cells) (Fig. 2f, g). The low levels of the basal keratins and the higher amounts of CD14 mRNA detected in cluster 2 and cluster 3 compared to cluster 1 (Supplementary Fig. S1b), indicate a luminal transcriptional profile of these clusters[22,23].

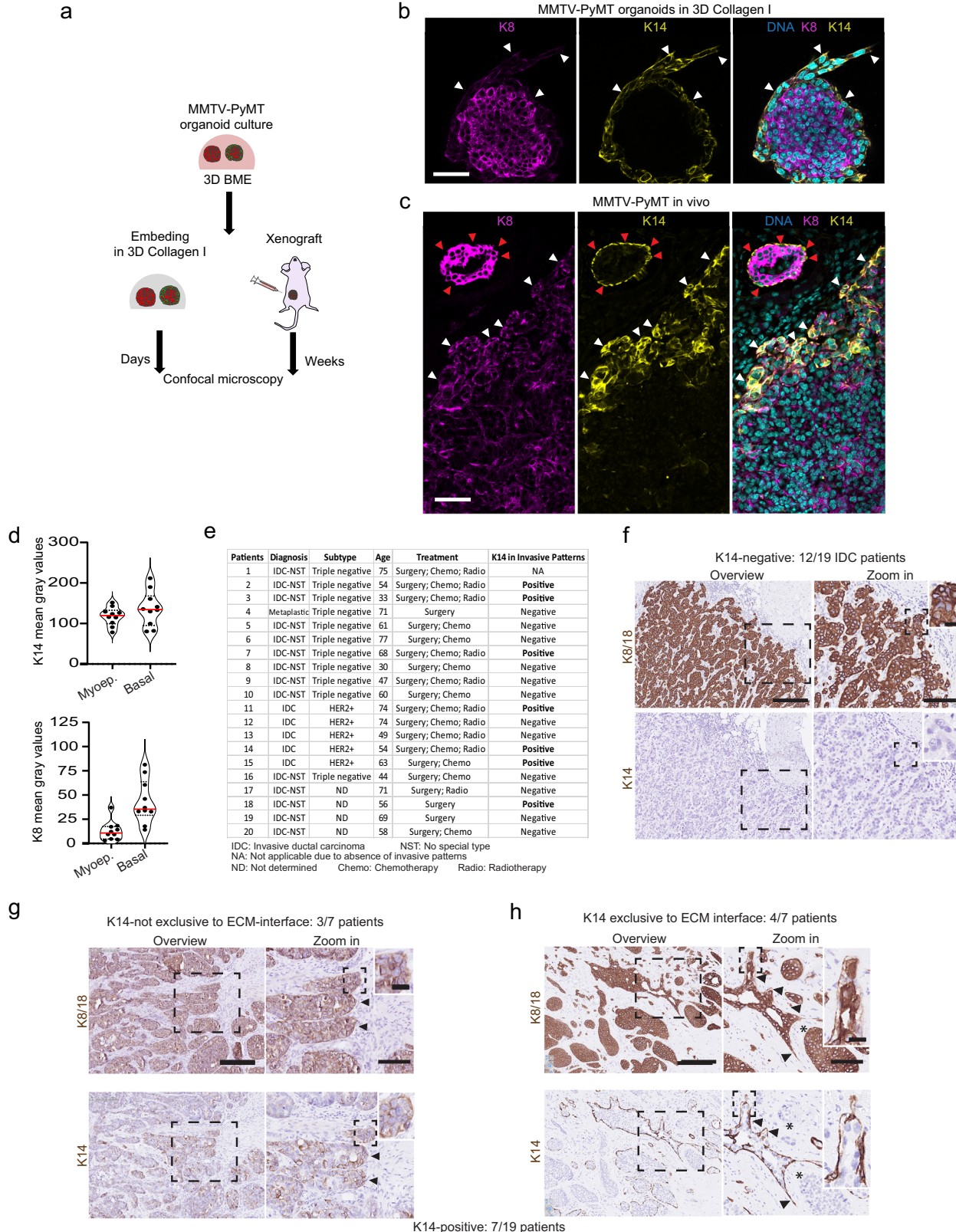

Taken together, our results show that Collagen I is neither required, nor sufficient, to induce the formation of basal like cells, an observation that was previously made in the same cellular model[24]. Yet, Collagen I drives a subpopulation of basal-like cells to become the protrusive leader cells of collective invasion. Thus, our model indicates that a basal-like cell identity is required for such collective invasion, while only few basal-like cells drive multicellular movement.

## A Yap-TEAD associated transcriptional program is activated in basal-like cells cultured in Collagen I

To understand if Collagen I induces changes in gene expression that induce invasion, we compared gene expression profiles between the basal-like cell populations in Collagen I and BME using the single cell mRNA sequencing data. When compared to BME, the basal-like cells from Collagen I showed higher expression of genes involved in ECM

**Fig. 1 | Basal-like cells form the invasive front in a subset of IDC patients and in MMTV-PyMT model in vitro and in vivo. a** In vitro and in vivo model based on MMTV-PyMT organoids cultured in 3D basement membrane extract (BME). After isolation from BME cultures, organoids were either embedded in 3D Collagen I or injected as cell suspension into the mammary fat pad of mice. **b, c** Confocal imaging of K8 and K14 in MMTV-PyMT organoids (**b**) grown in 3D Collagen I for 5 days or (**c**) in mammary fat pad for 13 weeks. **b, c** Representative images from three independent experiments. **b, c** White arrowheads: K14 positive cells located at the tumor-ECM interface and leading edges of the tumors; red arrowheads K14 positive cells lining mammary ducts. **d** Mean gray values of K14 and K8 in the K14 positive cells lining the mammary ducts (myoepithelial cells) and at the invasive tumor margins (basal-like cells). Red line median from 10 cells per cell type from one confocal slice from 1 experiment. **e** Summary of K14 expression in IDC. **f–h** Representative K8/K18 and K14 immunohistochemistry in serial tissue (whole) sections from IDC samples with zoom in on the cancer cell groups collectively invading the surrounding breast tissue. The invasive cancer cell groups showed (**f**) absence of K14 expression (12/19 IDC patients) (**g**) K14 expression throughout the invasive patterns (3/7 IDC patients) and (**h**) exclusive K14 expression in the cells located at the ECM-interface (4/7 IDC patients). Insets, K8/18 positive cells. Black arrowheads: K14 positive cells in the multicellular invasive cancer cell groups. Asterisk: K14 negative cells at the tumor-ECM interface. Scale bars: 250 (**f–h**), 100 μm (**c**), 50 μm (**b**; **f–h**, Zoom in), 10 μm (**f–h**, inset). Source data are provided as a Source Data file.

regulation (Fn1, Tnc, Col8a1, Col18a1, Loxl3), cytoskeletal dynamics (Vim, Tpm, Myh9), paracrine signaling (Inhba, Ctgf) and cell-cell signaling (Jag1) (Fig. 3a). Contrarily, basal-like cells from the BME samples expressed higher levels of genes related to the cell cycle (Pcna, Cdk4) and translation (Rpl3-5) (Fig. 3a and Supplementary Data 1). Basal-cell characteristics have been proposed to be induced by Collagen I, however except for Keratin 17 (K17), none of the major basal markers (K5, K14, Trp63) were upregulated in basal-like cells in Collagen I compared to BME (Supplementary Fig. S2a). Moreover, basal-like cells in Collagen I versus BME expressed similar relative levels of several classical EMT genes (*Snai2, Cdh2* and *Twist*) except for Fibronectin and Vimentin, which were upregulated by 6.7 and 2.08-fold in Collagen I compared to BME, respectively (Supplementary Figs. S1c–e and S2a).

To identify which transcription factors are differentially active in basal-like cells, we performed a motif analysis on the promotor regions of the differentially upregulated genes in collagen I. This analysis predicted elevated transcription factor activity in basal-like cells from Collagen I, with strong enrichment of the TEAD transcription factor family among the upregulated motifs (Fig. 3b). Subsequent gene set enrichment analysis revealed elevated activity of inflammatory pathways (Nfkb, Inflammatory response), EMT and Hypoxia in the basal-like cells from Collagen I. The basal-like population from BME showed higher activity in Oxidative phosphorylation, Myc signaling and cell cycle regulation (G2M, E2F) (Fig. 3c). Thus, Collagen I induces multiple transcriptional programs in basal-like cells that are distinct from the classical mesenchymal reprogramming previously implicated in breast cancer invasion.

Since Yap is a major TEAD-associated co-transcriptional activator, we tested if it is indeed activated in collectively invading cells. We analyzed the distribution of nuclear Yap by immunofluorescence, confocal microscopy and semi-automated in situ image analysis (Supplementary Fig. S2b). We focused on cells that are in direct contact with either BME or Collagen I (Supplementary Fig. S2b, 1st layer) and classified them as K14^HIGH or K14^LOW based on K14 expression (threshold: mean gray value). In BME, both K14^HIGH and K14^LOW rim cells showed low and comparable ratios of nuclear/cytoplasmic Yap (Fig. 3d, e). In contrast, K14^HIGH cells expressed significantly higher nuclear Yap levels in Collagen I compared to the K14^LOW rim cells (Fig. 3d, e). Nuclear Yap levels were further elevated in the invading K14^HIGH cells compared to the non-invading K14^HIGH cells at the organoid rim (Fig. 3d, white *versus* green arrowheads; Fig. 3f). The K14^HIGH cells leading the invasive strands showed increased nuclear enrichment of Yap, comparable to the breast cancer leader cells (Fig. 3d, Zoom ins, Fig. 3f). Unlike Yap, its close relative Taz was predominantly localized in the cytosol of K14^HIGH breast cancer leader cells as detected by immunostaining and confocal microscopy (Supplementary Fig. S2c) of MMTV-PyMT organoids using 3 different antibodies against Taz (Supplementary Fig. S2c, d). Finally, we tested whether YAP is also active in basal-like cells guiding collective invasion of human breast tumor cells. A limited number of human breast cancer organoids exhibiting K14 expression were identified, with none of the K14-negative samples displaying invasive characteristics in collagen I. The

K14-positive patient-derived breast cancer organoid (PDXO 1915)[25], similar to the MMTV-PyMT organoids, invaded in 3D Collagen I as cohesive multicellular strands guided by K14 positive cells enriched with nuclear Yap (Fig. 3g). In conclusion, using mouse and human breast cancer cell models, our data show increased Yap activity, specifically in the cells driving collective invasion in 3D collagen I.

## Yap is required for the emergence of leader cells and basal-guided ECM remodeling during collective invasion

To test the functional role of Yap in the emergence of leader cells and invasion, we interfered with Yap expression using doxycycline (dox)-inducible shRNA expression and with Yap-TEAD interactions by dox-inducible expression of Yap-Tead inhibitory peptide (YTIP)[26]. We scored the effect of interference on invasion in Collagen I. Dox-treatment caused a reduction in total Yap protein levels by 70-80% compared to control conditions (Supplementary Fig. S2e) and resulted in a significant reduction in the expression of classical Yap targets as well as Collagen I induced genes in the MMTV-PyMT mixed organoids (Supplementary Fig. S2f). Yap knockdown substantially reduced the number of protrusive leader cells, identified as the pointed tips of the invasive strands (Fig. 3h, arrowheads) and lowered the cumulative length of invasive strands by 50–90% after 3 days of culture in Collagen I (Fig. 3i, j). The inhibitory effect of Yap knockdown was also confirmed in freshly isolated MMTV-PyMT tumor organoids (Supplementary Fig. S2g, h). Similarly, collective invasion was reduced by at least 55% after dox-inducible expression of YTIP in MMTV-PyMT organoids (Supplementary Fig. S2i and Fig. 3k) and 4T1 breast cancer spheroids (Figure S2j, Fig. 3l). We next tested the effect of the Yap-Tead small molecule inhibitor (K975)[27] on the invasion of the breast cancer (PDXO 1915) patient-derived organoid model. Treatment of Collagen I cultures with K975 (5 μM) inhibited the mRNA expression of some of YAP-target genes *CYR61 and CTGF*, indicating inhibition of YAP transcriptional activity (Supplementary Fig. S2k). K975 treatment at concentrations of 5 and 10 μM reduced collective invasion of the breast cancer patient-derived organoids in Collagen I by at least 45% (Fig. 3m, n). Unlike the inhibitory effects on invasion, Yap knockdown or YTIP expression did neither affect the positioning of basal-like cells at the rim of the organoids (Fig. 4a) nor organoid growth rates, as assessed by measuring the change in the surface area of the organoid body (Supplementary Fig. S2l). Moreover, the ratio of basal-like cells that entered the cell cycle was not affected by YTIP expression. Confocal analysis shows that the ratio of basal-like cells with nuclear Ki67 were comparable between YTIP-expressing cells and controls (Supplementary Fig. S2m). Thus, Yap and its interaction with TEAD transcription factors are required for basal-like cells to emerge as leaders and function in driving collective invasion.

## Yap drives basal cell guided ECM remodeling

One of the key steps towards leader cell function is to interact with, pull and remodel adjacent Collagen fibers to induce their alignment and form the tracks for collective invasion[28]. Given the effects of Yap inhibition on leader cell function (Figs. 3h–n and 4a), we analyzed the

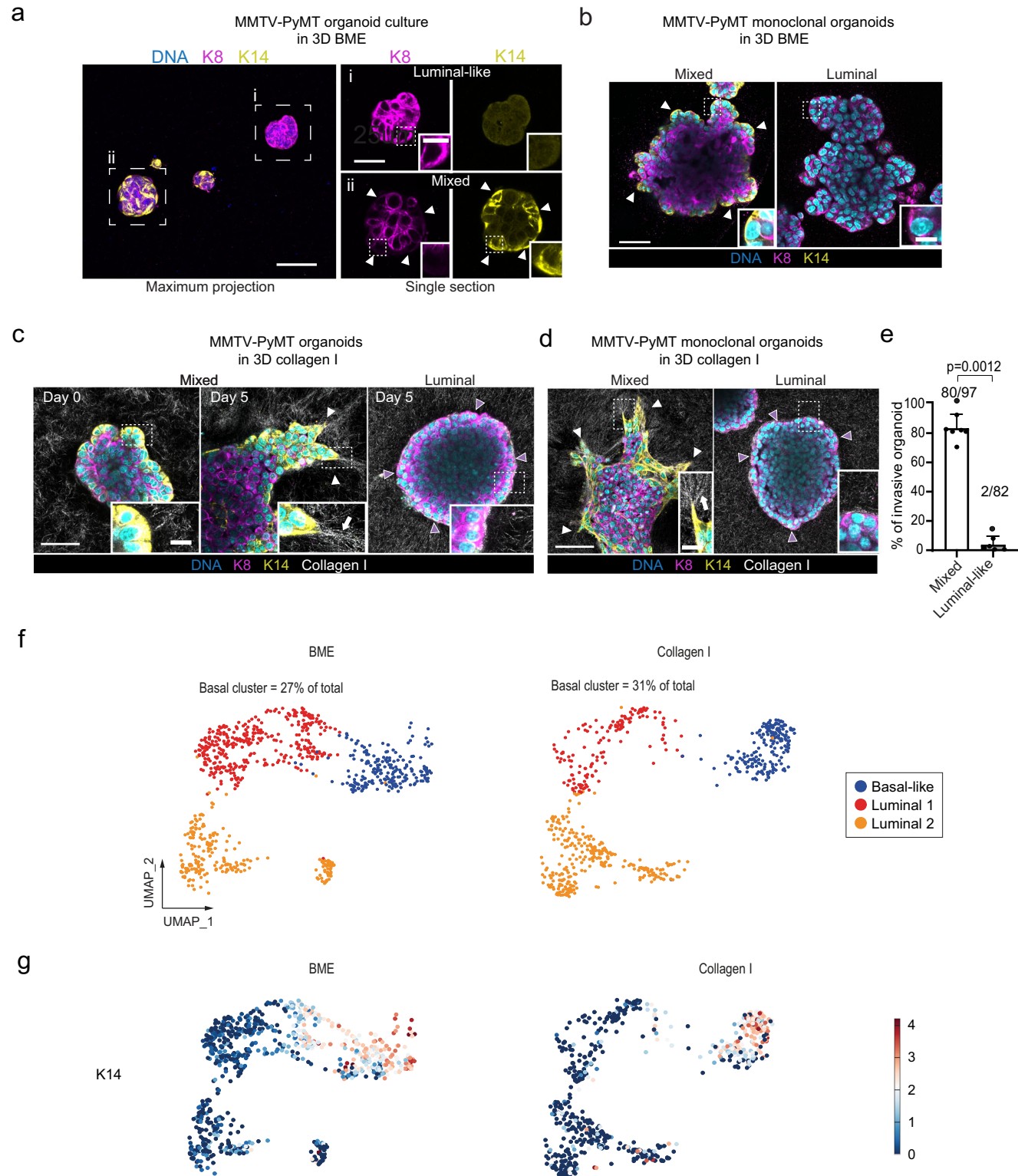

**Fig. 2 | Basal-like cells become invasive in Collagen I and drive collective invasion. a, b** Confocal imaging of Keratin 8 (K8), Keratin 14 (K14) in MMTV-PyMT (**a**) parental and **b** monoclonal cultures growing in 3D BME. a) Zoom in show the two subtypes of organoids (i) luminal and (ii) mixed. **a, b** Arrowheads indicate K14 positive (basal-like) cells located at the organoid-ECM interface. **c, d** Confocal imaging of MMTV-PYMT parental organoids embedded in 3D Collagen I (reflection). Protrusive K14-positive cells guide multicellular invasive strands. White arrowheads: leader cells, white arrows: Collagen I fiber alignment. Magenta arrowheads: non-protrusive K14-negative cells that are in contact with Collagen I. **a–c** Representative images from 3 independent experiments. **e** Percentage of monoclonal organoids showing at least 1 protrusive strand after 3 days in 3D

Collagen I (related to **d**). Average values from $n = 97$ (mixed) and $n = 82$ (luminal) organoids per condition from 7 (mixed) and 6 (luminal) independent experiments. Error bars, SD. *P* values, two-sided unpaired Mann–Whitney test. **f** UMAP plots based on transcriptional profile from mRNA sequencing of single cells isolated from MMTV-PyMT organoids cultured in 3D BME (799 cells from 3× 386-well plates) or Collagen I (627 cells from 3× 386 plates). Percentage of basal cells of the total amount of cells is indicated in the upper corner of the UMAP plots. **g** Plots of the same individual cells as (**f**) with the red/blue color-coded log2 cumulative read counts of basal marker K14. Scale bars: 50 µm (**a–d**), 25 µm (**ai**, **aii**), 10 µm (**a–d**, inset). Source data are provided as a Source Data file.

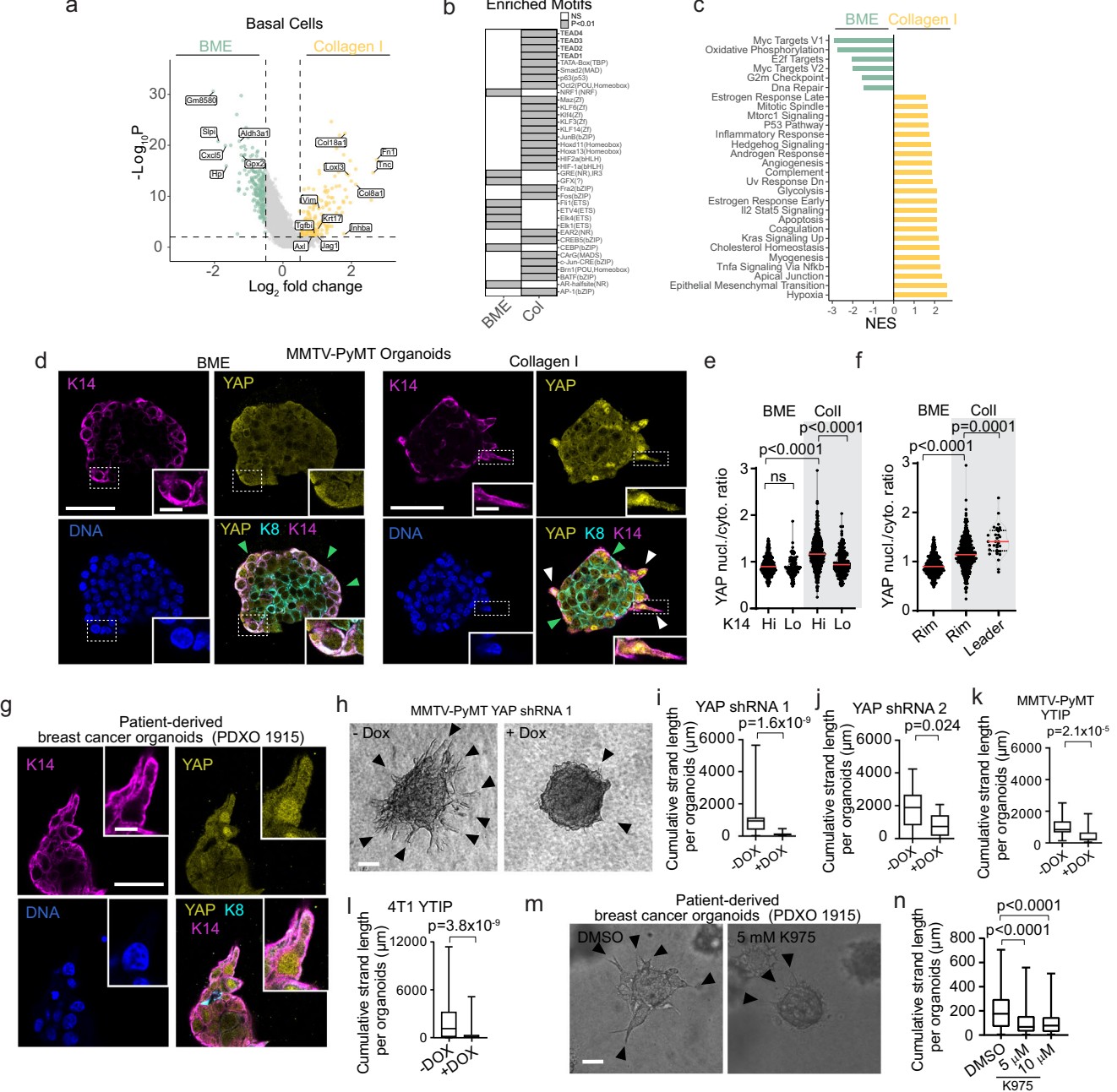

**Fig. 3 | Collagen I enhances a Yap-associated program in basal-like cells that is required to drive collective invasion. a** Differentially expressed genes in basal-like cells (Collagen I vs BME). *P* values, two-sided Wilcoxon rank sum test. **b** Enriched transcription factors (*P*≤0.01) based on motif analysis of the differentially expressed genes. *P* values, cumulative hypergeometric distribution without multiple comparisons adjustments (HOMER). **c** The significantly altered pathways from the gene set enrichment analysis, filtered for absolute values of NES >0.5 and enrichment *P* value (<0.01) produced by the GSEA. **d** Confocal imaging of Yap, K14 and K8 in MMTV-PyMT organoids embedded in BME/Collagen I (1 day). Arrowheads: K14-positive cells: rim (green); leader cells (white). **e** Yap nuclear/cytoplasmic ratio in cells located at the rim. K14-high (Hi) or -low (Lo), based on mean intensity. Red line: median from *n* = 407 (BME, K14 Hi), *n* = 92 (BME, K14 Lo), *n* = 627 (Collagen I, K14 Hi), *n* = 197 (Collagen I, K14 Lo) cells from three independent experiments. **f** Yap nuclear/cytoplasmic ratio in K14-high cells: rim (BME, *n* = 407; Collagen I *n* = 468) or leader cells (*n* = 33) from three independent experiments. **g** Yap, K14 and K8 in

patient-derived organoids Collagen I, day 3) from 2 independent experiments. **h** Yap shRNA (−/+ Dox) in MMTV-PyMT organoids (Collagen I). Arrowheads: invasive strands. **i–l** Cumulative strand length after (**i, j**) Yap knockdown or (**k, l**) YTIP expression in MMTV-PyMT organoids (Day 3) and 4T1 spheroids (Day 2) in Collagen I. **i–l** Medians (black line), 25/75 percentiles (boxes) and maximum/minimum values (whiskers) from *n* = 30 organoids from three independent experiments (**i, k**), *n* = 15 (-Dox) *n*=17 (+Dox) organoids from two independent experiments (**j**), *n* = 81 (-Dox) and *n* = 88 (+Dox) spheroids from three independent experiments (**l**). **m, n** Patient-derived organoids invading in Collagen I, DMSO, 5 µM or 10 µM of K975 (**n**) resultant collective invasion after 2–3 days in Collagen I, *n* = 112 organoids analyzed per condition from three independent experiments. **n** Representation similar to (**i–l**). *P* values, (**i–l**) two-sided unpaired Mann–Whitney test; **e, f, n** two-sided Kruskal–Wallis test (Dunn's multiple comparison). Scale bars: 50 µm (**d, e, h, m**), 10 µm (**d, g**, insets). Source data are provided as a Source Data file.

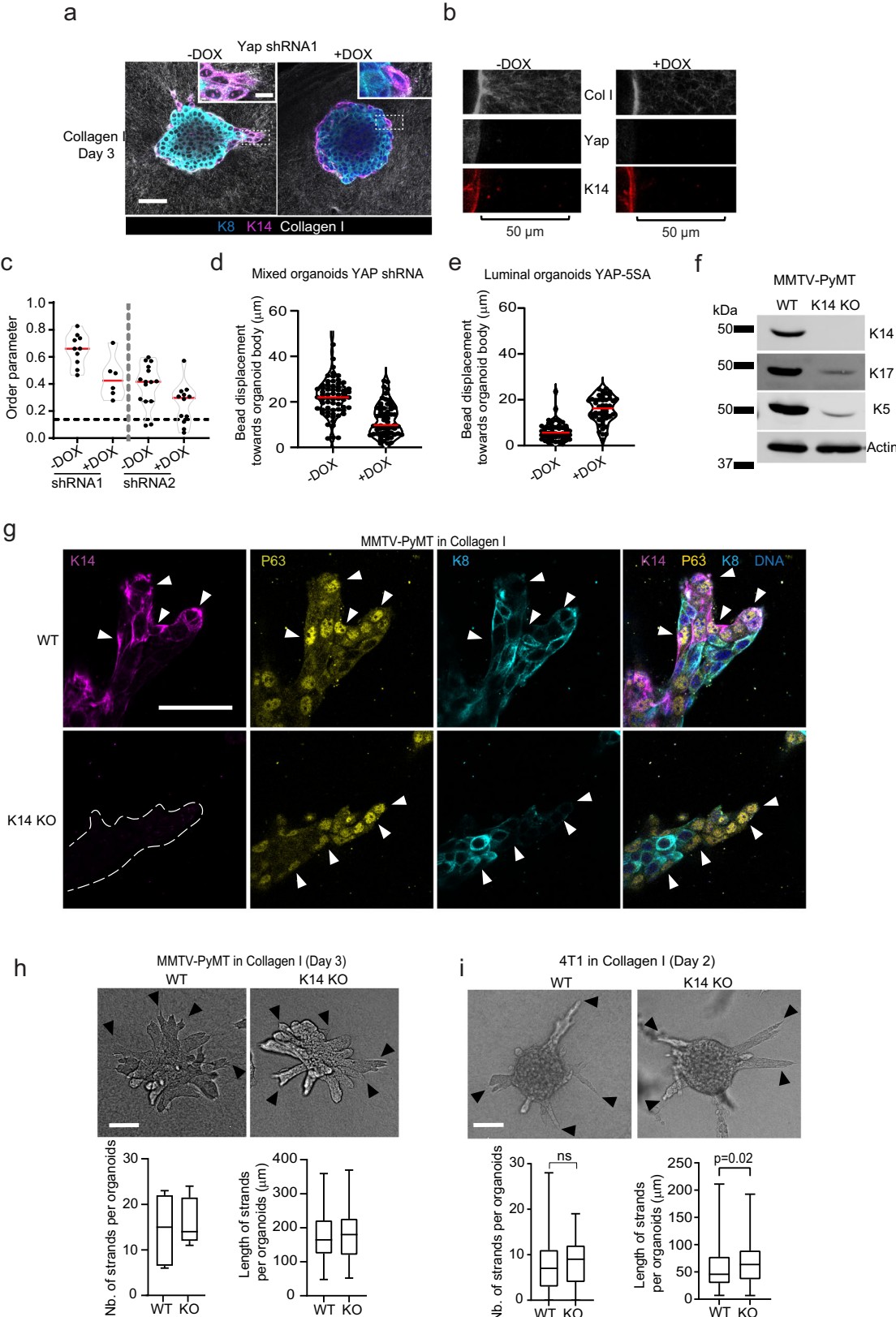

orientation of the Collagen fibers located within 50 μm of basal-like cells (at the rim) of the organoid by Orientation J plugin (Fiji) that uses an order parameter to quantify their alignment perpendicular to the organoid rim. Yap knockdown substantially reduced the alignment of the Collagen I fibers by basal-like cells (Fig. 4b, c). In addition to the Collagen alignment analysis, we embedded tumor organoids with

fluorescent beads in Collagen I followed by analyzing bead displacement because of the physical forces exerted by cells (Fig. 4d, e). After 6 h of culture, the average bead movement towards the organoid body was 21.79 μm (SD 7.854) and was reduced to 12.08 μm (SD 7.041) as a result of Yap knockdown (Supplementary Movie 3 and Fig. 4d). To further confirm the importance of Yap in ECM pulling, we

**Fig. 4 | Yap drives basal-cell guided ECM remodeling and collective invasion independent of K14. a** Confocal imaging of K14, K8, and Collagen I in control and Yap knockdown MMTV-PyMT organoids. Insets: K14-positive cells at the organoid-ECM interface. Representative image from three independent experiments. **b** Collagen I imaging using CNA35 GFP Collagen I dye in control and Yap knockdown organoids. **c** Collagen fiber order parameter near organoid rim (50 μm) with Yap shRNA1 or shRNA2 after 4 days of culture. Dotted line: average collagen order parameter within 50 μm from organoid rim on day 0 (mean = 0.16, SD = 0.08). Median (red line) $n = 10$ and $n = 6$ (shRNA1 -Dox and +Dox, respectively), $n = 15$ and $n = 12$ 6 (shRNA2 -Dox and +Dox, respectively). **b, c** Data from 1 experiment. **d, e** Bead displacement towards MMTV-PyMT organoids with dox-inducible (**d**) Yap shRNA1 mixed organoid after 6 h in Collagen I and **e** YAP 5SA luminal organoids after 39h in Collagen I (−/+ Dox). Median (red line) from $n = 63$ (-dox) $n = 60$ (+Dox) (**d**) and $n = 48$ per condition (**e**) beads from one experiment. **f** Western blot analysis from two independent experiments showing K14, K17, K5 and actin expression from whole cell lysates of wild-type (WT) and K14 knockout (KO) MMTV-PyMT organoid cultures. **g** Confocal imaging of K14, p63 and K8 in WT and K14-KO organoids invading in Collagen I for 3 days. Arrowheads: WT and K14-KO leader cells with high nuclear Yap and low K8 signals. **g** Representative images from $n = 8$ organoids from 1 experiment. **h, i** WT and K14-KO (**h**) MMTV-PyMT organoids and **i** 4T1 spheroids invading Collagen I. Arrowheads: invasive strands. **h, i** Number and length of strand in WT and K14-KO organoids, medians (black line), 25/75 percentiles (boxes) and maximum/minimum values (whiskers) from (**h**) 72 (WT) and 84 (KO) invasive strands from five organoids per condition from one experiment and **i** 87 spheroids per condition from three independent experiments. *P* values, two-sided unpaired Mann–Whitney test. Scale bars: 50 μm (**a, g**), 100 μm (**h, i**), 10 μm (inset: **a**). Source data are provided as a Source Data file.

overexpressed a constitutively active form of Yap (5SA, contains 5 Serine-to-Alanine mutation) in luminal organoids. DOX-induced YAP activation promoted pulling on the ECM, reflected by the 2.5 times increase in the movement of beads towards the organoid body compared to control luminal organoids that showed minimal bead displacement (6.244 μm, SD 4.109) even after 39 h of contact with Collagen I (Supplementary Movie 4 and Fig. 4e). Thus, Yap is required for basal-like cells to remodel and align Collagen I fibers.

## Collective invasion is independent of K14 expression

K14 has been functionally implicated in leader cell function as well as in the regulation of Yap via its interaction with 14-3-3S[29]. To directly investigate its importance in Collagen I induced collective invasion, we performed CRISPR/Cas9 mediated deletion of the K14 gene in the mixed MMTV-PyMT tumor organoids. Deletion of *Krt14* was confirmed by DNA sequencing (Supplementary Fig. S3a) and subsequent western blot analysis showed absence of K14 protein (Fig. 4f). In line with previous reports[30], *Krt14* deletion caused a strong reduction (> 80%) of the basal keratins K5 and K17 (Fig. 4f). K14 KO organoids still contained basal-like cells as detected by the high expression of p63 (basal-cell marker) (Fig. 4g, arrowheads) and low K8 levels (Supplementary Fig. S3e, arrowheads). KO of K14 was not sufficient to prevent organoid invasion (Fig. 4h). Similar to MMTV-PyMT tumor organoids, K14 KO in 4T1 breast cancer cell line (Supplementary Fig. S3c) did not affect collective invasion in Collagen I (Fig. 4i). Despite K14 KO, the basal leader cells showed a protrusive morphology and maintained nuclear enrichment of Yap (Fig. 4g–i and Supplementary Fig. S3e, arrowheads). We therefore conclude that Collagen I-mediated Yap activation and collective invasion do not require K14 expression.

## Tension on Collagen I fibers contributes to the Yap-driven transcriptional program in leader cells

Since Yap is a mechano-transducer, we investigated whether Collagen-I mechanics cause Yap activation in basal-like cells. Detaching Collagen gels (after polymerization) from the underlying surface of the culture plate is an established method to perturb mechanics of the collagen network[31,32]. Detached/floating gels cannot sustain tension buildup and consequently, cells fail to establish or maintain Collagen I fiber alignment[33].

First, we assessed the tumor organoid phenotype in static conditions and observed that most MMTV-PyMT organoids failed to invade in floating Collagen gels (Fig. 5a, b) and the transition of basal-like cells into leader cells was strongly inhibited (Supplementary Fig. S3b). We also found that, while Yap mRNA levels were similar in organoids embedded in floating and attached Collagen gels, the expression levels of some classical and Collagen I-induced Yap targets were lower in floating conditions (Fig. 5c). These data confirm that Yap-mediated transcription is dependent on tension in Collagen I fibers.

Next, we addressed the role of Collagen I biomechanics in controlling Yap activity in leader cells. First, we observed that a partial digestion of the Collagen I network using short term treatment with a low concentration of collagenase (3 min, 0.5 mg/mL) abolished nuclear localization of Yap in both rim and leader cells compared to control conditions (denatured collagenase) (Supplementary Fig. S3d). Although it highlights the strong dependence of Yap activity on Collagen I biomechanical conditions in both low (rim cells) and high (leader cells) Yap activity cells, it does not allow us to understand the distinction between the two populations. As visible in Supplementary Movie 1 and Supplementary Movie 2, the Collagen network remodels dynamically to establish areas where fibers align perpendicular to the tumor organoid edge, which is caused by organoid-Collagen interactions and subsequent pulling forces. This is supported by previous reports showing that dynamic pulling forces are applied on Collagen I fibers by the leading edges and essential for collective invasion of breast and colon cancer cells[28,33]. Based on these observations, we hypothesize that the high mechanical tension present in aligned Collagen fibers[34] is responsible for the increased activity of Yap in leader cells. To directly test this, we allowed tumor organoids to become invasive in Collagen I gels for 24 h and then detached the gels from the surface of the dish to perturb the high tension. In detached gels (2 h 30 min), Yap nuclear levels in leader cells (still in contact with aligned Collagen I) dropped to levels comparable to the non-protrusive basal-like rim cells (Fig. 5d, e). In contrast, the non-protrusive basal-like cells (rim cells) showed similar Yap nuclear levels in both floating and attached gels (Fig. 5d and Supplementary Fig. S3b). Thus, Yap activity in leader cells is specifically inhibited by reducing ECM tension, strongly suggesting that mechanical forces on aligned Collagen I fibers are essential for the increased Yap activity in leader cells.

Since Yap is first required for basal-like cells to align Collagen I fibers and further activated by ECM tension, our work proposes a YAP-centered and mechanically regulated feed-forward mechanism that drives collective cancer invasion.

## Yap promotes local invasion in vivo and is expressed in invading strands in patient breast cancer samples

To test whether Yap is required for collective tumor invasion in vivo, we injected MMTV-PyMT organoids harboring dox-inducible Yap shRNAs into the 4th mammary fat pad of mice (Fig. 6a). After tumors reached a volume of 50 mm³, mice were randomized, and Yap knockdown was induced by supplying dox-containing food. Dox-treatment did not affect tumor growth as monitored by the increase in the tumor volume over time, except for the slight increase in tumor volume observed as of week 11 in the dox-treated mice (Fig. 6b). We next performed histological analysis of the tumor borders by calculating the protrusive index based on the perimeter and the convex hull ratio of the tumor border (Supplementary Fig. S4a). From these measurements, Yap knock-down inhibited the protrusions of the tumor by 23% compared to control tumors, whose invasive borders were rich in multicellular strands extending into the mammary stroma (Fig. 6e, f). We then tested the impact of Yap-Tead inhibition (YTIP) on

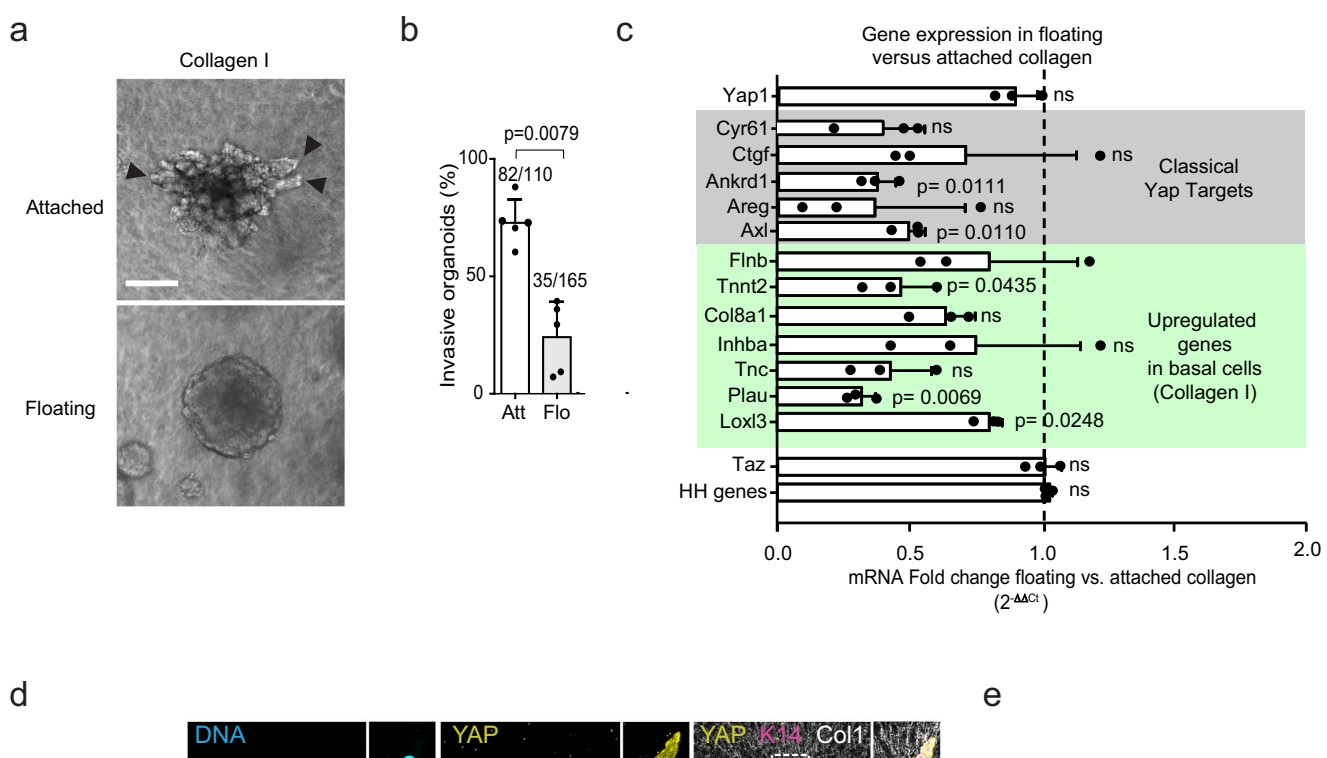

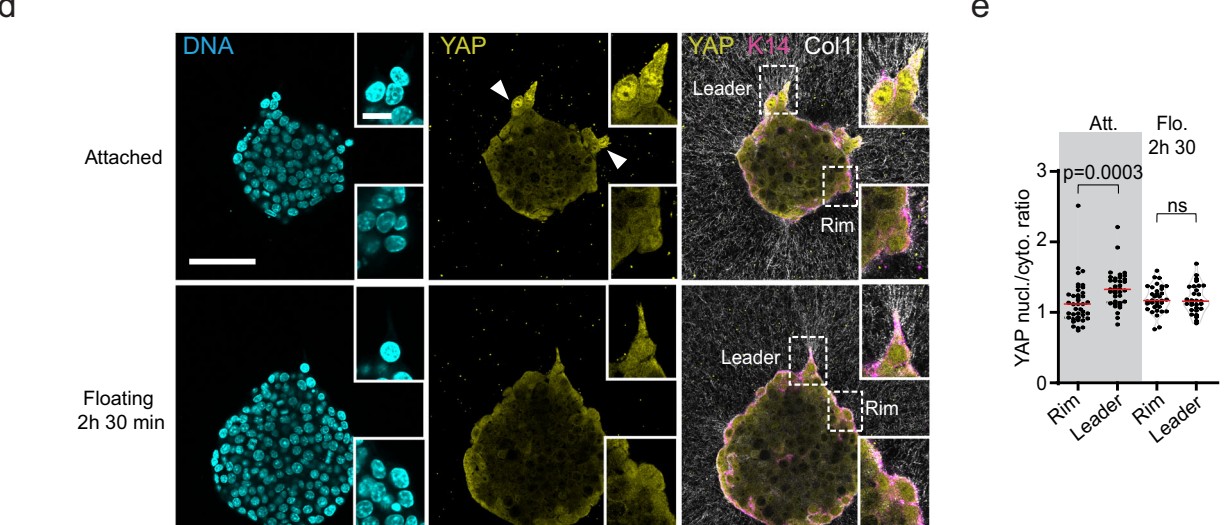

**Fig. 5 | Collagen I induced forces control Yap-mediated transcription and invasion. a** Representative bright-field images of MMTV-PyMT organoids embedded in attached (Att.) and floating (Flo.) Collagen I gels. Arrowheads: invasive strands. **b** Percentage of organoids that have 1 or more invasive strands (invasive organoids). Data show average values with SD from $n = 110$ (Att.) and $n = 165$ (Flo.) organoids from five independent experiments. *P* values, two-sided unpaired Mann–Whitney test. **c** qPCR data showing relative mRNA expression of the classical Yap targets and other genes (identified from single cell sequencing) in MMTV-PyMT organoids embedded in floating relative to attached Collagen I gels. The bar graph represents the average values with SD from 3 independent experiments. *P* values, two-sided Student's *t* test. **d** Yap distribution in rim and leader cells after detaching the Collagen gels for 2 h 30 min. **e** Yap nuclear/cytoplasmic ratio in the K14 positive cells that are either located at the rim or at leader positions in attached and floating gels. Data are derived from 29 to 40 cells per condition from three independent experiments. *P* values, two-sided Kruskal–Wallis test with Dunn's multiple comparison. Scale bars: 50 μm (**a**, **d**), 10 μm (**d**, inset). Source data are provided as a Source Data file.

tumor growth and invasion in vivo. Tumor organoids harboring dox-inducible YTIP showed a comparable increase in volume over time (Supplementary Fig. S4b). Similar to Yap shRNA1, organoids expressing dox-inducible YTIP show less protrusive morphology (by 19%) (Fig. 6d, g, white arrowheads) and more rounded borders (Fig. 6d, red arrowheads) compared to control. In addition to the primary tumors, we analyzed the number and relative size of the spontaneous metastasis formed in the lungs of the control and dox-treated mice (Yap shRNA1, 2 and YTIP). Our analyses show no significant differences between dox-treated mice and controls, in both the number and size of metastases in the lungs (Supplementary Fig. S4c). Based on our

invasion analysis, we show that Yap is a major regulator of collective invasion in 3D Collagen I in vitro as well as in the complex mammary tissue surrounding primary tumors in vivo.

To compare our findings to human breast cancer, we analyzed YAP expression in IDC samples. We first focused on the subset of patients that showed basal-like cells in their invasive patterns. Immunohistochemical analysis showed that Yap expression was enriched in the nucleus of the K14 positive cellular subsets (Supplementary Fig. S4d, e, arrowheads). By confocal microscopy and in situ image analysis we found that ~60% of the tumor cells with nuclear Yap expressed K14 (Fig. 6i, j). A stronger association was observed between

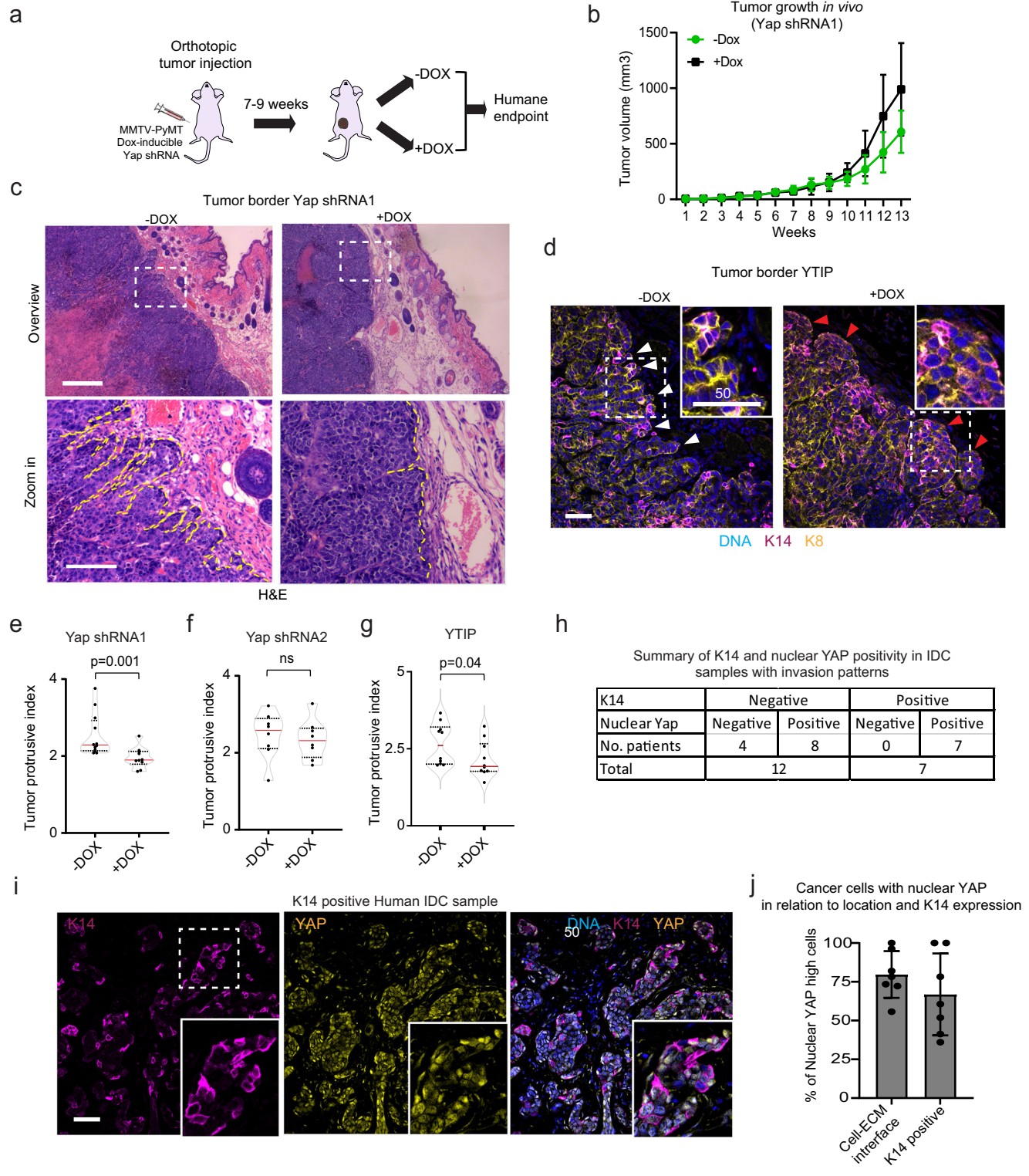

**h** Summary of K14 and nuclear YAP positivity in IDC samples with invasion patterns

| K14 | Negative | | Positive | |
|---|---|---|---|---|
| Nuclear Yap | Negative | Positive | Negative | Positive |
| No. patients | 4 | 8 | 0 | 7 |
| Total | 12 | | 7 | |

nuclear Yap and cell position relative to the ECM, as ~75% of the cancer cells with nuclear Yap are located at the tumor ECM interface (Fig. 6j). Interestingly, invasive IDC cells expressing nuclear YAP were also detected in 8 out of 12 of the K14 negative invasive tumors (Supplementary Fig. S3f). This indicates that K14 positivity associates with but is not essential for Yap activation in breast cancer cells, consistent with our K14 KO experiments in 3D Collagen I in vitro. The evidence of YAP activation in collectively invading cancer cells in patient samples underscores the likely translational relevance of the Yap-centered leader cell emergence mechanism uncovered in this study.

## Discussion

We here uncover a feed-forward loop between Collagen I and Yap signaling in basal-like tumor cells as a key driver for collective breast cancer invasion. Yap activation is initiated by Collagen I contact and subsequent feed-forward induction of a transcriptional program that promotes ECM remodeling leading to local alignment of Collagen I. Establishment of aligned Collagen organization through activated basal-like cancer cells leads to the emergence of protrusive leader cells that propel collective invasion, the predominant mode of dissemination in breast cancer (Fig. 7).

**Fig. 6 | Yap promotes breast cancer invasion in vivo. a** Experimental strategy for injecting MMTV-PyMT organoids harboring dox-inducible Yap shRNA1 or shRNA2 in the 4th mammary fat pad. **b** Tumor volumes were monitored weekly using digital caliper in the context of doxycycline (dox)-containing or control food with the starting time of dox treatment (week 7 or 9) depending on tumor size. Values represent the average tumor volume with SD. 13 mice (2–13 weeks) per condition. **c** Representative hematoxylin and eosin (H&E) stainings of primary tumors from control and dox-treated mice. Dashed lines indicate the invasive front contours. **e–g** quantification of tumor invasion assessed by protrusive index (related to Supplementary Fig. S3a) in response to Yap knockdown (shRNA1 or shRNA2) and expression of YTIP. **d**, **e** Each value represents the average of (**e**, **f**) at least 2–4

regions of tumor borders (10× objective) per tumor from $n = 12$ (-Dox) and $n = 10$ (+Dox) for shRNA1 and $n = 8$ mice per condition for shRNA2 and **g** 1–3 regions (20x objective) per tumor from $n = 10$ mice per condition. P values, two-sided unpaired Mann–Whitney test. **h** Summary of K14 and nuclear Yap protein expression in IDC samples. **i, j** K14, and Yap immunostaining of (whole) section from a K14 positive IDC sample (Representative image $n = 7$ tissue sections). Zoom ins depict multicellular invasive cells with nuclear YAP distribution, particularly in the basal-like cells located at the interface with the breast tissue. **j** Percentage of cells with SD that have nuclear Yap that are located at the interface with the ECM and that are K14 positive. Data represent average percentage from 7 K14 positive IDC patients. Scale bars: 100 μm (**c**), 50 μm (**d, i**). Source data are provided as a Source Data file.

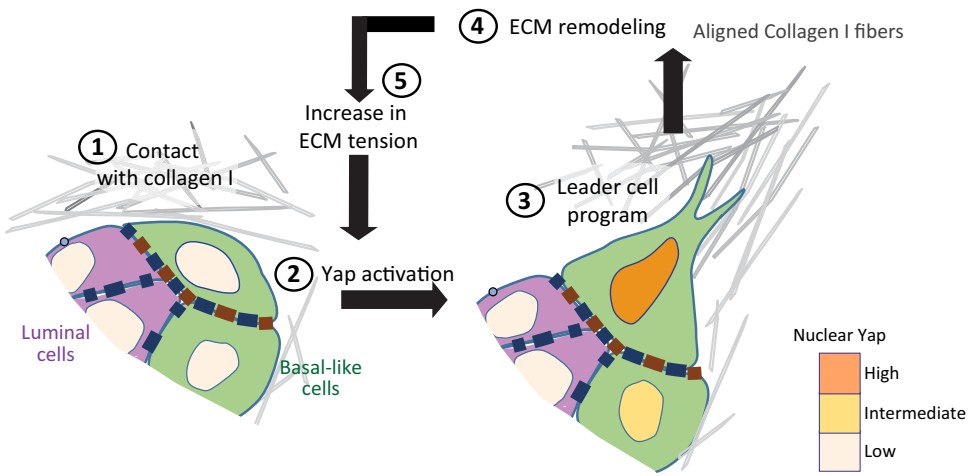

**Fig. 7 | Bidirectional Yap-mediated cell-ECM interactions drive the emergence of leader cells and collective invasion.** Proposed sequential steps (1–5) underlying the induction of basal cell driven collective invasion by Collagen I. Biomechanical cues within Collagen I fibers are mechanotransduced by basal-like cells through Yap, which activates a transcriptional leader cell program. The leader cell program

induces basal-like cells to protrude into and remodel the Collagen I-rich ECM. The remodeled ECM is mechanotransduced by basal-like cells through further Yap activation creating a feed-forward loop that initiates and maintains protrusive collective invasion.

Collagen I interactions have been proposed to promote invasive characteristics and establishment of leader cells by activating luminal to basal cell reprograming. The expression of K14, which is associated with basal reprograming, was functionally implicated in the induction of collective invasion by Collagen I in MMTV-PyMT breast cancer organoids[1,35]. In our model system, a basal-like transcriptional program was not induced by the interaction of the organoids with Collagen I, as we did not observe basal-like cell emergence from fully luminal organoids, and because comparable percentages of basal-like cells were observed in mixed organoids grown in BME and Collagen I. Therefore, we focused on elucidating the mechanisms promoting the transition of pre-existing basal-like cells into invasive leader cells, rather than mechanisms driving luminal to basal cell reprogramming[1,36].

The K14-independent invasion that we observe in 3D collagen I contradicts previous reports where K14 knockdown inhibits invasion of the MMTV-PyMT tumor organoids[1]. Cheung et al., used lentiviral transduction of freshly isolated tumor organoids followed by rounds of antibiotic selection to enrich for cells that express the shRNA. Whether the K14 KD organoids still contain basal-like cells after the rounds of selection remains unclear. We show by p63 immunostainings that the basal-like cells are still present in the K14 KO organoids and able to drive collective invasion. We cannot exclude compensatory mechanisms (because of gene KO and culturing) that allow basal-like cells to drive invasion independent of K14, K5 and K17. In patients, collective invasion occurs without the expression of K14 in many

clinical breast cancer samples, in line with previous reports that show K14-negative IDC tumors using immunohistochemical analysis and qPCR[37]. Altogether, our work show that leader cells can still emerge and drive invasion without the expression of K14.

Similar to luminal to basal reprograming, epithelial to mesenchymal transition (EMT) can also be induced by Collagen I interactions[38,39]. In our model, basal-like cells express higher levels of some EMT-associated genes compared to luminal cells. However, in collagen I, we did not observe the induction of many classical EMT genes (Snail, Twist, Cdh2), except for Fibronectin and Vimentin, suggesting partial EMT characteristics acquired by the basal-like leader cells. Indeed, invasive basal-like leader cells maintain E-cadherin expression with the follower cells during collective invasion in collagen I[40]. Thus, our studies show that Collagen I promotes a transcriptional program that is different from luminal to basal cell reprograming and classical EMT.

Whereas most of the reported oncogenic functions of Yap involve activation of cell proliferation and stem cell properties[13,14], we show that Yap mediated transcription induces basal-like breast cancer cells to become protrusive and lead collective invasion. In vitro, YAP knockdown caused a strong inhibition of invasion, however, in the in vivo setting, the inhibitory effects of YAP knockdown on invasion was less pronounced. The influence of the complex tumor microenvironment in vivo might explain the differences in the extent of invasion inhibition compared to the in vitro models. The presence of

additional factors in vivo, such as multiple ECM components, stromal cells and paracrine factors, may contribute to the insufficiency of Yap inhibition in tumor cells to strongly block invasion, unlike in simpler 3D models. Future investigations should aim to elucidate the Yap downstream targets that are essential for invasion and metastasis, which will clarify its potential as a therapeutic target.

ECM remodeling is a major determinant for the establishment of leader cells and the drive of collective invasion[41,42]. We show that Yap-driven collective invasion depends on ECM remodeling. Conversely, progressive Collagen alignment near basal-like cells at the organoid rim is lost upon perturbance of Yap. Collagen I alignment is strongly dependent on mechanical pulling mediated by the cytoskeletal actin-rich protrusions[34]. Several of the genes that are upregulated in basal-like cells (incl. Vim, Tpm2) regulate actin protrusions and may contribute to cytoskeletal changes that accommodate Collagen I pulling and alignment[43–45]. In addition, Yap controls the expression of ECM components and ECM-modifying enzymes (incl. Tnc, Col8a1, Loxl3 and Plau), several of which have been shown to support cancer cell invasion and associate with poor clinical outcome in breast cancer patients[40,46–48]. Transcriptional effects on ECM remodeling place Yap at the center of a previously unappreciated mechanical feed forward loop in the induction of leader cell functions and collective invasion. This could explain the stochastic occurrence of invasive strands at the tumor/organoid ECM interface where the ECM becomes stiffer because of remodeling including Loxl3-mediated crosslinking[40]. Patches of aligned and stiffened Collagen I activate YAP and lead to the transcription of ECM remodeling factors in basal-like cells at the tumor-ECM interface. Collagen I structure changes further leading to larger areas of Collagen alignment and build-up of mechanical tension that further increases Yap activity in the adjacent cells to become the leaders of invasion. Indeed, in IDC, basal leader cells have been shown to guide the invasive strands almost exclusively toward regions with aligned Collagen fibers[1,40]. The molecular mechanism underlying Collagen I-induced activation of Yap in basal-like cells remains to be elucidated. Mechanical activation of Yap in leader cells may depend on cell-ECM receptors (incl. integrins, DDRs)[49–51], cell-cell junctions between leader and follower cells[40,52], and on mechanically activated channels (incl. Piezo, connexin hemichannels)[6,53,54] that are proposed in several models to be active in the cytoskeletal protrusions of leader cells.

The well-described mouse models used in our study recapitulate the organization of many human breast tumors containing luminal and basal-like cells. However, of the many human breast cancer organoid lines investigated, we only found very few that contained basal-like cells. Using human breast cancer organoid line (PDXO 1915), we show the importance of YAP signaling for basal-like cells to become leaders of invasion in human breast tumor. Further validation of the wider relevance awaits the entry of additional basal-like cell-containing human breast cancer organoid lines in the field.

In conclusion, we uncovered a mechanism in which Yap regulates a transcriptional program that is driven by reciprocal ECM interactions. The resulting signals induce basal-like cancer cells to orchestrate collective breast cancer invasion.

## Methods

Our research complies with all relevant ethical regulations and approved by The Netherlands Food and Consumer Product Safety Authority (NVWA) of the ministry of Agriculture, Nature and Food, and the Tissue Science Committee of the University Medical Center Utrecht.

### Antibodies and reagents

The following primary antibodies (Supplementary Table 1) were used: mouse anti-human YAP (Cat# sc-101199, Santa-Cruz), rabbit anti-human YAP (Cat# 4912, CST), rat anti-mouse Keratin 8 (Cat# 531826, DSHB), rabbit anti-human Keratin 14 (Cat# 905301, Biolegend), mouse anti-human Keratin 14 (Cat# ab7800, Abcam), rabbit anti-human Keratin 5 (Cat# ab52635, Abcam), rabbit anti-human Keratin 17 (Cat# ab183330, Abcam), rabbit anti-human TAZ (ab84927, Abcam), mouse anti-chicken Actin (Cat# 1501, Millipore). The following secondary antibodies were used: Alexa Fluor-488/568/647-conjugated goat anti-mouse, -rabbit or -rat antibody (Invitrogen). Odyssey goat anti-mouse 800 and goat anti-rabbit 680. To visualize the nucleus and F-actin, 4′,6-diamidino-2-phenylindole (DAPI, Thermo Fisher) and Alexa Fluor-568-conjugated phalloidin (Thermo Fisher) were used, respectively.

### Organoid and cell culture

MMTV-PyMT breast cancer organoids were a generous gift from Jacco Van Rheenen[17]. The breast cancer patient-derived organoids PDXO 1915 were obtained from the Park Lab (McGill University, Canada)[25]. As characterized in ref. 25, the PDXO 1915 organoids (GCRC1915Tc) are derived from an IDC-NOS female breast cancer patient with basal molecular subtype. The patient had distant metastases and was treated with adriamycin/doxorubicin, cyclophosphamide, carboplatin and taxol/paclitaxel. The 4T1 metastatic breast cancer cell line was used (CRL-2539 ATCC).

Tumor organoids (MMTV-PyMT, PDXO 1915) were cultured in 20–40 μL drops of 3D Cultrex PathClear® RGF basement membrane extracts type 2 (BME)[55]. Similarly, 4T1 cells were treated as the organoids and cultured in 3D BME where they grew from single cells into multicellular spheroids. MMTV-PyMT PDXO 1915 tumor organoids as well as 4T1 spheroid cultures were maintained (37 °C, 5% $CO_2$, humidified atmosphere) in FCS-free culture medium. For both MMTV-PyMT and 4T1 cells, the media consisted of Advanced DMEM/ F12 media containing 10 mM HEPES (Cat# 15630-056, Life Technologies), 1× Glutamax (Cat# 35050-038, Life Technologies), 100 Units/mL Penicillin-Streptomycin (Cat# DE17-602E, Westburg), 1.25 mM N-acetyl-cysteine (Cat# N0636, Sigma-Aldrich), 1× B27 (Cat# 17504-044, Life Technologies), 2.5 nM bFGF2 (Cat# 100-26, PeproTech), 100 μg/mL Primocin (Cat# ant-pm-05, InvivoGen). PDXO 1915 were cultured in medium prepared from Advanced DMEM/F12 reduced (12634028, Gibco) with addition of 1% penicillin-streptomycin (pen/strep, 15070-063, Gibco), 1% GlutaMAX™ supplement (35050-061, Gibco), 1% HEPES (15630080, Lonza). Medium was also supplemented with 1.25 mM N-Acetyl-L-Cysteine (NAC; A9165-5G, Sigma), 1× B27 (17504001, Gibco), 50 μg/mLPrimocin (ant-pm1, Invivogen), 100 ng/mL Noggin (120-10C-50UG, Peprotech), 250 ng/mL R-spondin-3 (120-44-100UG, Peprotech), 10 mM Nicotinamide (N0636-100G, Sigma), 5 ng/mL FGF-7 (100-19-10UG, Peprotech), 20 ng/mL FGF-10 (100-26-25UG, Peprotech), 500 nM A83-01 (SML0788-5MG, Sigma), 37.5 ng/mL Neuregulin (100-03-50UG, Peprotech), 500 nM SB202190 (S7067-5MG, Sigma) and 5 ng/mL hEGF (AF-100-15-1MG, Peprotech). For passaging[55], tumor organoids and spheroids were collected by disrupting the BME drops mechanically followed by incubation with 300–500 μL of trypsin-EDTA (TrypLE Express, Cat# 12605-010) for a maximum duration of 1 min at 37 °C. When the organoids and spheroids are dissociated into single cells and small clusters, trypsin digestion was inhibited by the addition of trypsin inhibitor (Cat#: T9003, Merck Sigma), followed by a washing step and centrifugation (314×$g$, 5 min). Cell pellets were suspended in BME. After polymerization, the gels were supplemented with culture media. All organoids and cell lines were negative for mycoplasma contamination (MycoAlert Mycoplasma Detection Kit, Lonza).

For the generation of monoclonal cultures, MMTV-PyMT organoids were dissociated into single cells and seeded at a density of 1 cell/well in a 96-well plate containing BME. Wells containing doublets, clusters or more than one cell were disregarded. Only wells (8/96) with clear and morphologically distinct individual cells were kept and followed over a period of 5–7 days. In total, 4/8 of the single cells grew

into organoids and 3 of these monoclonal organoids were propagated and analyzed for K14 and K8 expression and invasion.

## Lentiviral production and tumor organoid transduction

The following Yap shRNA sequences Yap shRNA1: CCGGTGAGAACAAT GACAACCAATACTCGAGTATTGGTTGTCATTGTTCTCATTTTT; Yap shRNA2: CCGGGAAGCGCTGAGTTCCGAAATCCTCGAGGATTTCGGAA CTCAGCGCTTCTTTTT were inserted into the pLV FUTG Tetracycline Inducible shRNA vector using the In-fusion cloning kit (Clontech). Similarly, YTIP-GFP cDNA was inserted into the pInducer20 vector[56]. Insertion of the Yap shRNA oligos in the pLV FUTG and YTIP-GFP in pInducer20 were confirmed by sequencing using the following primers FUTG-tet Ind-shRNA GGGTTTATTACAGGGACAGC and GATCA CTCTCGGCATGGACG, respectively.

For forced expression of constitutively active YAP1 (YAP1-5SA), Myc-YAP-5SA from plasmid pQCXIH-Myc-YAP-5SA (Addgene plasmid #33093) was integrated in a pDONR221 backbone and the Myc-tag was replaced with a Flag-tag (cloning done by Epoch Life Science). Base-pairs 5'-CAATGCGGAATATCAATCCCAGCACAGCAAATTCTCCAAAAT GTCAGG-3' were added between base pairs 982 and 983 of the YAP gene similar to the previously described YAP constructs[57]. Insertion of the Flag-tag and base pairs were confirmed by sequencing with primers 5'-CCACCATGGACTACAAGGACGA -3' and 5'-CTGGGTCGACTATAA CCATG-3'. The Flag-YAP1-5SA was inserted into a pInducer21 backbone with Gateway cloning, to generate pInducer21- FLAG-YAP1-S5A-Ubc-rtTA-IRES-GFP x pLV. Successful integration was confirmed by sequencing with primers provided by the Gateway cloning kit (pInducer-Fwd and pInducer-Rev).

To generate lentiviral particles HEK293 cells were transfected with the FUTG-tetInd-shRNA, pInducer20-YTIP-GFP or pInducer21-FLAG-YAP1-S5A using X-tremeGENE™ (Sigma Aldrich)[58]. Supernatants containing viruses from HEK293 cultures were harvested 48 and 72 h post-transfection and concentrated by ultra-centrifugation for 2.5 h at 175,000×$g$ (Optima LE-80K Ultracentrifuge). MMTV-PyMT organoids were incubated for 4 h with concentrated viruses supplemented with 4 μg/mL Polybrene (Sigma Aldrich). Selection for the transduced organoids was performed 3 days after infection by supplying the organoid culture media with 10 μg/mL blasticidin (Cat# ant-bl-05, Invitrogen) or 2 μg/mL puromycin. Culture medium was refreshed every other day for at least 10 days. After selection, organoids were treated with Doxycycline (Cat# D9891, Sigma-Aldrich) (2 μg/mL) for 3 days and knockdown efficiency was assessed by qPCR, western blot analysis and florescent microscopy.

## Generation of K14-deficient organoids and organoids with sfGFP-tagged K14

K14-deficient MMTV-PyMT organoids and 4T1 cells were generated with a pX458 vector[59] that expresses Cas9-T2A-GFP and the K14 sgRNA gAAGGGCTCTTGTGGTATCGG targeting the first coding exon. To this end, 5 μg of pX548 vector was mixed with 200 μL Optimem and 12 μL TransIT transfection reagent (MIrus) and incubated for 15 min at room-temperature. Hereafter, this mix was added to 2.8 ml of a freshly trypsin-treated single cell suspension in complete organoid medium. After 48 h, GFP-positive cells were sorted using a FACS Aria II flow cytometer (BD) and plated in a six-well plate. The next day cells were trypsinized and embedded in BME. Up to 20 single organoids were picked from BME cultures using a Gilson P20 pipette under an inverted microscope, trypsizined separately and expanded to generate mono-clonal organoid cultures. They were checked by western blotting for the absence of K14. Genetic ablation of K14 was further confirmed by sequencing. Genomic DNA from organoids was isolated using the QIAamp DNA Micro Kit (Qiagen). The region around the Cas9 target site was amplified in a 50 μL reaction containing 1 μL Pfu polymerase (Promega), 5 μL 10× Pfu polymerase buffer (Promega), 3 μg isolated gDNA, 200 nM forward primer Krt14 Fwd: CCTCCAGCTAAGT

GCCAGTC and 200 nM reverse primer Krt14_Rev: GTCCATTTGACC-CACCTTGC and 100 nM dNTPs each (Roche) in PCR-grade dH₂O. The PCR program was as follows: initial denaturing at 95 °C for 2 min, denaturing for 1 min at 95 °C, annealing at 60.8 °C for 1 min, extension at 72 °C for 1 min and final extension at 72 °C for 5 min. The middle three steps (denaturing, annealing and extension) were repeated 30 times. PCR products were isolated form TAE agarose gels, isolated using QIAquick Gel Extraction Kit (Qiagen) and send for sequencing at Macrogen using either the Krt14_Fwd or Krt14_Rev primer.

C-terminal tagging of K14 was done by scarless tagging via non-homologous end-joining. For this purpose, three constructs were used. Construct 1 (SL-1097_krt14_lin1_krt14_1) drives expression of an sgRNA (gCAACCGCCAGATCCGCACCA) targeting the last coding exon of K14 and contains the donor-fragment. This fragment contains all K14 amino acids downstream of the double-stranded break but upstream of the K14 stop-codon in frame with sfGFP and a P2A-puromycin selection marker. It was generated by cloning the sequence CCACCA CCAAAGTCATGGATGTGCACGATGGCAAGGTGGTCTCCACCCACGA GCAGGTCCTGCGCACCAAGAACGGGGGGTCTGGTGGCAGTGGAGGG in between the NotI and BamHI site of SL-1097 (kind gift of Prof. S. Lens and M. Vromans, UMCU, Utrecht). Construct 2 encodes Cas9 and the sgRNA gCACATCCATGACTTTGGTGG, which targets construct 1 upstream of the donor fragment. Construct 3 expresses Cas9 and an sgRNA GGCCAGTACCCAAAAAGCGG that targets construct 1 down-stream of the donor-fragment. Constructs 1, 2 and 3 were transfected in a ratio of 2:1:1 by the method described above. Hereafter, cells were transferred to BME and after 3 days, selected with 2 μg/mL puromycin for 9 days. Several surviving green-fluorescent organoids were hand-picked under an inverted epifluorescence microscope, separately trypsinized and re-embedded into BME to obtain monoclonal cultures. For each of these clones, K14-GFP endogenous tagging was confirmed by sequencing using the following primers K14 Fwd: GGAGTGA GGTGGTAAACGGG.

GFP Rev: GCCAAGGTACCGGCAGTTTA. In addition, protein expression of the GFP-tagged K14 was confirmed by confocal micro-scopy and western blot analysis (Data not shown).

## 3D BME and Collagen I assays

Organoids and spheroids were isolated from BME cultures using 1 mg/mL of Dispase II (Cat# 17105041, Life Technologies) followed by three washes with DMEM media. Organoids were either embedded in BME or in type I Collagen lattices consisting of non-pepsinized rat-tail Col-lagen I at a final concentration of 2.0 mg/ml[55]. Briefly, Collagen mixture was prepared containing 10× phosphate buffer saline (PBS) (Gibco), NaOH and dH₂O according to the manufacturer's instructions. Col-lagen was pre-polymerized for 2–3 h on ice. Afterwards, organoids were embedded in the Collagen I mixture followed by incubation at room temperature for 5 min and plating as 45 μL drops on a pre-warmed 24-well plate (Costar) and incubated at 37 °C for polymeriza-tion. After polymerization, complete organoid medium was added.

Manipulation of Collagen I biomechanics was performed by detaching the gel from the well, rendering it floating[55]. For partial digestion of the Collagen I network, collagenase (Cat# C0130, Sigma-Aldrich) was supplemented to the media at a final concentration (0.56 mg/mL). As a negative control, denatured form of collagenase was used by boiling for 10 min. Collagen gels were incubated with collagenase and denatured collagenase for 3 min at 37 °C, followed by at least 4 washes with culture media and incubation for 30 min (37 °C, 5% CO₂, humidified atmosphere). After that, samples were fixed with 4% paraformaldehyde (PFA; Sigma-Aldrich).

## Bright-field microscopy

Organoids embedded in BME or Collagen I were imaged using EVOS M5000 microscope (2× objective NA = 0.08, 10x objective, NA=0.4) at various time-points. For quantification of invasion, the number of

pointed and protrusive tips (leader cells) per spheroid was counted manually, as well as length of invading strands[60], using the image analysis software ImageJ (ImageJ; 1.40v; National Institutes of Health).

## Immunofluorescence of 3D in vitro cultures

For immunostaining, organoids embedded in BME or Collagen I were fixed with 4% paraformaldehyde (PFA; Sigma-Aldrich) for 10 min at room temperature and washed three times with 1× PBS. Fixed samples were blocked using 10% normal goat serum (Gibco) in 0.3% Triton-X (Sigma-Aldrich) for 1 h. Primary antibody was diluted in antibody buffer (0.3% Triton-X with 1% w/v BSA in 1× PBS) using the following antibody dilutions Yap (1:100), Taz (1:25–1:100), K14 (1:300), K5 (1:100), K17 (1:200), K8 (1:50); P63 (1:50). Gels were incubated with the primary antibody for at least 20 h at 4 °C with shaking, followed by at least 4 washing steps with 1× PBS (10–15 min each). Samples were then incubated with Alexa Fluor-488/568/647-conjugated secondary antibodies (1:500) together with DAPI (5 μg/mL) shaking overnight at 4 °C. Organoids were imaged using Zeiss LSM 880 (40× objective, NA=1.1).

## Quantification of Yap levels

Mid z-section of the organoids imaged by confocal microscopy were analyzed for nuclear and cytosolic Yap levels using a customized program in MATLAB R2018a. This program selects organoids from background by intensity thresholding on the combined K8, K14 signal. It detects nuclei based on intensity thresholding and k means clustering on the Dapi signal and segments cells using a watershed on the detected nuclei and multi-scale hessian aggregation. The program automatically characterizes the cells as being in contact with the Collagen ('outer') or not ('inner') (Supplementary Fig. S2b). Background-subtracted mean intensity values in the nucleus and cytosol are used to calculate the ratio of nuclear over cytosolic Yap levels. Leader cells vs rim cells are selected manually based on the following criteria. Rim cell is a cell located at the first layer and in contact with the ECM. Leader cells is an elongated and protrusive cell that extend outward relative to center of the organoid. For the floating (2 h 30 min) and attached collagen experiments, manual selection of K14 positive leader cells and rim cells was performed. Floating for 2 h 30 min did not affect the elongation of leader cells, assessed by measuring the length of leader cells based on K14 + F-actin signal (Average length of leader cells +/− SD: Attached; (16.7 μm +/− 6.45); Floating: 18.76 μm +/− 5.63) Collagen I conditions. The nuclei and cytosols of both leader and rim cells were selected manually, based on Dapi and K14 + F-actin signal, respectively, using Fiji. After background correction, nuclear Yap level per cell (leader or rim) was calculated by diving the nuclear over cytosolic Yap mean gray values.

## Analysis of Collagen I alignment

In order to quantify Collagen alignment, single confocal slices of Collagen I (labelled with GFP-CNA35, Collagen I probe generated in house) were analyzed using a publicly-available ImageJ plugin, OrientationJ, as previously described[40]. OrientationJ determines the local orientation (from −90° up to 90°) and coherency (from 0 to 1) of every pixel relative to its neighbor in rectangular regions (30 × 50 μm) of the Collagen I that are perpendicular to the cell-ECM border. The zero distance was set at the cell–matrix interface determined by the F-actin or K14 signal. The nematic order parameter was calculated from the second-order tensor order-parameter S2 using the orientation measurements, S.

$$S_2 = \begin{bmatrix} \langle cos2\theta \rangle & \langle sin2\theta \rangle \\ \langle sin2\theta \rangle & -\langle cos2\theta \rangle \end{bmatrix} \tag{1}$$

<.> denote averages of all orientation measurements. Solving the eigenvalue problem for $S_2$ yields two eigenvalues that is the scalar order-parameter S familiar for liquid crystals in 2D:

$$\lambda_{1,2} = \pm S = \pm \sqrt{\langle cos2\theta \rangle^2 + \langle sin2\theta \rangle^2} \tag{2}$$

The scalar order parameter enumerates the distribution of orientation measurements. When the scalar order parameter is 0 then distribution of angles is uniform. When the scalar order parameter is close to 1 then the distribution of angles is sharply-peaked. Similar order parameter calculations have been used previously to analyze the orientation of other biological networks such as fibrin[61] and F-actin[62].

## Analysis of bead displacement

To assess the role of Yap in ECM pulling, MMTV-PyMT mixed and luminal organoids expressing, respectively, dox-inducible Yap shRNA or constitutively active form of YAP were cultured in BME with or without DOX for two days before embedding in Collagen I. Both organoid lines were collected from BME using Dispase (as described above) followed by embedding in Collagen I (2 mg/mL) together with fluorescent beads (FluoSpheres® carboxylate-modified microspheres, 1.0 μm, red fluorescent (580/605), Molecular Probes, Oregano, USA) at a dilution of 1 μL of beads in 600 μL of Collagen I in an Ibidi LabTek imaging slide. The MMTV-PyMT mixed organoids (Yap shRNA 1) and the luminal organoids (5SA) were imaged immediately or after 4 days of culture in Collagen I, respectively, in the presence or absence of dox. This was done by confocal time-lapse microscopy (Zeiss, LSM880, 20X, NA = 0.75) at 37 °C and 5% CO₂. Imaging of the mixed organoids (Yap shRNA 1) was performed with an interval of 37 min for 6 h whereas the luminal organoids were imaged for a longer period (39 h) with an interval of 10 min. Drift correction (xy) was performed for affected time-lapse images was done using the "Correct 3D Drift" plugin in Fiji. The beads that are located within an organoid's diameter (mixed organoids) or radius (luminal organoids) distance from each organoid border were selected for quantification of displacement towards the organoid body. Beads that were located in between two close organoids or that were not trackable throughout the whole time-lapse were excluded from the analysis.

## Immunohistochemistry and immunofluorescence of tumor tissues

Immunohistochemistry (IHC) and immunofluorescence were performed on 4–5 μm whole invasive breast cancer tissue sections diagnosed as high grade invasive ductal carcinoma (IDC). We selected high-grade Luminal A, Luminal B, and Triple Negative invasive breast cancers to cover the invasive spectrum as complete as possible (Fig. 1e). The use of material was approved by the Tissue Science Committee of the University Medical Center Utrecht and informed consent was obtained from the participants. IHC procedures were performed standardized on the Roche Ventana™. In short, after antigen retrieval by boiling for 20 min in 10 mM Citrate pH 6.0 or Tris/EDTA pH 9.0 (depending on the used primary antibody), a cooling period of 30 min preceded the 1 h primary antibody incubation. Hematoxylin was used as a counterstaining. K8 and K14 antibodies were provided optimized and ready to use by Roche. For YAP IHC we used 1:50 dilution (Santa Cruz 63.7 SC-101199). K8, K14 and Yap IHC were analyzed using the Nanozoomer Digital Pathology Software (NDP 2.6.17) and K8/18 positive invasive cell groups that contained at least 10% of the cells with clear signal for cytosolic K14 or nuclear Yap were scored as positive.

For immunofluorescence, antigen retrieval was performed using Tris-EDTA buffer (15 min, 95–100 °C) followed by immunostaining as previously described[60]. Briefly, tissue sections were blocked (5% normal goat serum in TBST) for 1 h at room temperature followed by primary antibody incubation at 4 °C (overnight) using the following antibody dilutions: Yap (1:100), K14 (1:300), K8 (1:50). After antibody incubation, samples were washed four times (TBST, 5 min each) and incubated with secondary Alexa Fluor 488/546/647-conjugated goat

anti-mouse, anti-rabbit, anti-rat antibodies and with DAPI (1 μg/mL), for 1 h at room temperature. Then, samples were washed four times (TBST, 5 min each) and mounted with Immu-Mount (Epredia).

## Western blot analysis

Organoids in BME and Collagen gels were lysed by placing the gels in SDS sample buffer (62.5 mM Tris-HCl, 2% SDS, 10% glycerol, 50 mM DTT and bromophenol blue), followed by sonication for five cycles at 4 °C (Bioruptor). Whole cell lysates were boiled for 5 min, loaded, and separated on a 10% acrylamide gel with Tris-glycine running buffer followed by blotting to PVDF membrane by wet transfer. PVDF membranes (Cat# IPFL00010, Millipore) were blotted for actin (1:2500) or GAPDH (1:5000) as loading control; YAP (1:1000); K14 (1:1000); K17 (1:500), K5 (1:1000) followed by fluorescence detection (Typhoon) and image analysis using the software (Image studio Lite ver. 4.0).

## Real-time qPCR

For RNA isolation, Collagen gels containing cancer organoids (MMTV-PyMT and PDXO 1915) were collected and lysed by mechanical disruption in 500 μL TRIzol™ reagent (Thermo Fisher) per gel. RNA was isolated using the Qiagen RNeasy® Lipid Tissue Mini Kit. 500–1000 ng RNA was converted into cDNA using the iScript™ cDNA Synthesis Kit (Bio-Rad)[55]. In each reaction tube, cDNA, and Fwd and Rev primer concentrations of 0.4 μM were used. qPCR primers had a product length between 50 and 150 bp and confirmed to specifically bind the mouse gene of interest using the NCBI n-BLAST tool. Primers were validated using cDNA titration and were approved when $R^2 > 0.98$, amplification efficiency was between 80-120% and when only one clear melting peak was generated during the melting curve analysis of the qPCR program. The primers used for qPCR are listed in Supplementary Table 2. qPCR was performed using FastStart™ SYBR® Green Master reaction mix, using a thermal cycler (Applied Biosystems). Data were analyzed using the CFX Maestro software, and fold changes were calculated using the ΔCt method.

## Single-cell mRNA sequencing

Organoids cultured in BME or Collagen I for 5 days were isolated from the gels using combination of Dispase II and collagenase. Isolated organoids were dissociated into single cells by trypsin-EDTA and plated as single cells in 384-well plates (3 × 384-well plates per ECM condition) using FACS sorter (FacsCalibur). Single-cell mRNA-sequencing was performed by Single Cell Discoveries (Utrecht, the Netherlands) according to published protocols[63]. Count matrices were imported in Seurat v4 for downstream analysis[64]. After removal of ERCC spike-in RNA and cell filtering (<10% mitochondrial reads, <8000 nFeatures, >500 nCounts), batch effects from different sorting plates were removed using reciprocal PCA. Dimension reduction was performed using UMAP based on the first 30 principal components and subsequent clustering was done with shared nearest neighbor (SNN) modularity optimization. Upon inspection of mitochondrial percentage and number of expressed features, one additional cell cluster was removed due to poor quality. The remaining cells were visualized with UMAP and clustered with SNN (resolution = 0.1). The basal identity of the cell clusters was evaluated with a module score of basal genes (i.e., *Krt5/14/17, Cdh3, Trp63, Itga6*). Differentially expressed genes were identified using FindMarkers and pathway activity was evaluated with a gene set enrichment analysis on the Hallmark gene sets from the Molecular Signature Database[65]. Transcription factor motif analysis was performed with HOMER on the promotor regions of the differentially expressed genes (logFC > 0.5, P < 0.01) between the groups of basal-like cells[66].

## Orthotopic injection of MMTV-PyMT organoids in mice

Recipient female nude mice (Female Athymic Nude- Foxn1nu, 6-9 weeks, Envigo) were housed at 20–24 °C, 45–65% humidity and Dark / light cycles (from 7 pm to 7 am lights off, from 7 am to 7 pm lights on). Mice were orthotopically transplanted in the 4th (inguinal) mammary gland with MMTV-PyMT cells harvested from organoid cultures in BME. We injected 250,000 cells in 50 μL serum free DMEM-F12 medium (Gibco). Mice were anesthetized using isoflurane (IsoFlo; Le Vet Pharma). Burprenorphine (0.1 mg/kg) was injected subcutaneously as analgesic treatment. After a recovery period of 2 weeks, tumor growth was measured using a digital pressure-sensitive caliper (Mitutoyo) on a weekly basis. Treatment started when tumors reached a volume of 50 mm³. Mice were randomized and either kept on their regular diet or switched to doxycycline containing chow (200 mg/kg; A155D70201; Ssniff, Bio services) ad libitum. Mice were euthanized if mammary tumor reached a size of 1000 mm³ or when mice showed severe discomfort. The maximal tumor size/burden was not exceeded.

Primary tumors were harvested, snap frozen and sectioned (4 μm). Sections were stained by H&E (Yap shRNA1 and 2) or immunostained for K8 and K14 (YTIP) and 1–4 regions of the tumor border per sample were imaged using Zeiss Axioskop 40 (10x objective, NA= 0.25) or confocal microscope (Zeiss, LSM880, 20X, NA = 0.75). The H&E images (Yap shRNA1, 2) and confocal images (YTIP) were analyzed by an automated macro (Fiji) that identifies the borders of the tumor by applying Gaussian blur, thresholding, and binary filters (erosion). This allows the creation of the perimeter selection of the tumor border and generation of the convex hull. The tumor protrusive index was then calculated by dividing the perimeter of the tumor border by the convex hull (Supplementary Fig. S4a). For lung metastasis, H&E sections of the lung were analyzed by at least two researchers and scored for the number of tumor foci and percentage of tumor composition relative to the lung tissue per section using Slide Score B.V. The averages of the scores per sample were used for comparison between treated and control groups.

## Studies approval

All animal experiments were performed in accordance with local, National and European guidelines under permit AVD1150020209964 issued by The Netherlands Food and Consumer Product Safety Authority (NVWA) of the Ministry of Agriculture, Nature and Food. The use of patient material was approved by the Tissue Science Committee of the University Medical Center Utrecht and informed consent was obtained from the participants. Banking of human specimens and associated clinical data is approved by MUHC research ethics board (study approval SUR-2000-966 and SUR-99-780). All patient data and biological samples were obtained from patients at the MUHC after obtaining informed consent.

## Statistical analysis

Graph Prism (version 8.0) was used for statistical analysis. The two-tailed unpaired Mann–Whitney (two groups) or Kruskal–Wallis test (more than two groups) with Dunns' correction were used except for the qPCR and Collagen alignment data where Student *t* test was performed after confirming normal distribution of the data by D'Agostino and Pearson omnibus normality test.

## Reporting summary

Further information on research design is available in the Nature Portfolio Reporting Summary linked to this article.

## Data availability

The single-cell sequencing data generated in this study have been deposited in the Gene Expression Omnibus database under accession number GSE197821. Due to the sizes of the files, the raw data from confocal microscopy, immunohistochemical analysis and qPCR were not provided and will be given upon request from the corresponding author. Source data are provided with this paper.

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

## Acknowledgements
This work was sponsored by grants from The Netherlands Organization for Scientific Research (NWO-TOP 02007), the European Union's Horizon 2020 FET Proactive program under the grant agreement No. 731957 (MECHANO-CONTROL), and Dutch Cancer Society KWF 2020-13552. The breast tissue and data bank at McGill University is supported by the Réseau de Recherche en Cancer of the Fonds de Recherche du Québec-Santé and the Québec Breast Cancer Foundation and certified by the Canadian Tumor Repository Network (CTRNet). We thank Dr. Gerard van der Krogt, Livio Kleij, Ingrid Verlaan, Marjolein Vliem, Cynthia Frederiks, Apostolis Nikolakopoulos, Colinda Scheele, Laura Bornes, Mojtaba Amini and Mathijs Baars for technical support.

## Author contributions
A.A.K., P.W.B.D., and J.d.R. designed the study. A.A.K., D.S., P.H., T.K., K.A.J., K.v.Z., L.J., L.E., M.V.d.N, A.G.E.G, D.V., M.P., M.T., D.W., and M.G.R. performed the experiments. D.S. and M.P.V. performed bioinformatics analysis. L.E and K.v.Z. performed the in vivo experiments. A.A.K. D.S., P.H., K.A.J., L.E., F.J.T.Z., R.F., S.T., and M.G. analyzed the data. A.F., S.T., M.P., P.v.D., and C.M. established and provided the patient-derived organoids. A.A.K., P.W.B.D., and J.d.R. wrote the manuscript. All authors read the manuscript and were given the opportunity to provide input.

## Competing interests
The authors declare no competing interests.
