## [Peer Review File · Nature Communications]

REVIEWER COMMENTS

Reviewer #1 (Remarks to the Author):

In this manuscript, Khalil and colleagues describe the existence of a feedforward loop between ECM composition/stiffness, YAP activity and ECM remodeling in a distinct subset of doubly K8 and K14 positive cells derived from the PYMT mouse breast cancer model. This proposed loop enables these cells to acquire a "leader cell" behavior and to promote collective cell migration in the presence of collagen.

Main points:

(1) The authors describe phenotypes that are different from similar experiments made by others, with apparently the same cell system. While this is not surprising per sé, it would be important to resolve these differences or at least highlight the possible reasons. For example, the authors use organoids derived by others and (presumably) kept for long time in culture. This might have introduced biases due to initial bottlenecks, or to selection in vitro. So, also in light of the reproducibility of the data, it would be important to repeat key experiments based on freshly-isolated PYMT cells.

(2) While PYMT have been for a long time a model to study breast cancer, in reality they do not recapitulate any of the cancer phenotypes/classes that are observed in human breast cancers. So one important issue is whether these observations are specific to the PYMT model, or more general. This issue is also relevant in light of the recent <https://www.nature.com/articles/s41467-021-27220-9> where it was shown that the type of ECM, rather than mechanical properties, regulate K8/K14 in human patient-derived breast cancer spheroids.

(3) The authors show that K14+ve basal-like cells preferentially activate YAP in response to collagen, which promotes their "leader cell" invasive action. However, K14 itself is not required for such preferential activation of YAP and "leader cell" phenotype. Based on single-cell sequencing, can the authors identify some other marker of "K14+ve" basal-like cells to ensure that the leader cells emerging in K14-KO organoids are of the same lineage, and different from the other "luminal only" cells?

(4) The authors discuss that their data uncover a previously unappreciated feedforward loop between ECM mechanical properties, YAP activation and further ECM remodeling. First, this is not entirely new as many observed previously similar loops in many cell types. This needs to be acknowledged. Second, this is not really demonstrated, apart from the acquisition of a leader cell morphology/position. There is no demonstration of a YAP-dependent ECM remodeling. For example, one could measure traction forces at the organic interface, and show these are higher facing K14+ cells, and absent when YAP is inhibited. Moreover, there is no demonstration this specifically occurs in leader cells. One way of demonstrating this could be to engineer cells to specifically inhibit YAP when they become K14+, or to mix TAM-treated K14+ cells with K14^{low} cells without shRNA from a homogeneous organoid. Moreover, is this a cell intrinsic ability of K14+ cells observed also when they are single? Are these more contractile because of YAP?

Minor points:

- the authors define their 3D cultures "organoids". Are these containing other cell types besides breast cancer cells? If not, maybe it would be better call them "spheroids" as more commonly used in the cancer biology field.

- in the initial characterization, it should be made clear (perhaps, also with an IF on a normal mammary duct) whether all PYMT cells are K8+ve and a subset ALSO expresses K14, and whether

this is different from the normal K8/K14 dichotomy observed in the normal duct.

- line 119 what are the non-invasive conditions? BME without added collagen? Please make clear in the text when first describing.

- bold and underlined genes in 3a should also include *INHBA* and *CTGF* that are recurrent YAP/TAZ-dependent genes in many differential expression analyses.

- are K14^{LOW} rim cells in Collagen also displaying nuclear YAP compared to the same cells in BME or not? This would be important to know whether basal-like cells are insensitive to collagen-mediated YAP activation, or just less sensitive. This could be enriched by also seeding single cells on top of BME vs collagen, to see whether such differential sensitivity relates to the multicellular environment or to intrinsic differences.

- what is the specificity of the TAZ antibody?? The result in S2C is unexpected, but TAZ antibodies are often non-specific and it is important to make the difference.

- the effects of Yap inhibition on leader cell and organoid proliferation should be pinpointed better by using more specific markers (EdU incorporation for example). In the context of collective migration proliferation does play a role in "pushing" cell strands and it would not be contradictory to show this.

- in vivo data need to be repeated with a second shRNA for YAP or with an independent way of inhibiting YAP, or with a rescue experiment. Any shRNA could have off-target effects on cell migration or any other phenotype, and the specificity of the YAP shRNA is unknown.

Reviewer #2 (Remarks to the Author):

In this study, using the MMTV-PyMT mammary carcinoma organoids, Khalil et al. showed that Collagen I drives a force-dependent loop, specifically in basal-like cancer cells, for collective breast cancer cell invasion. The feed-forward loop is centered around the mechanotransducer Yap and independent of K14 expression. Yap promotes a transcriptional program that enhances Collagen I alignment and tension, which further activates Yap. Active Yap is detected in invading breast cancer cells in patients and required for collective invasion in 3D collagen I and in the mammary fat pad of mice. The most note-worthy work is the finding of a new function for YAP in leader cell selection during collective cancer invasion. Although there some new findings, which are interesting, many data are quite descriptive and some further studies are needed.

1. Their single-cell mRNA sequencing results showed that the proportion of basal-like cells was significantly lower in collagen I than in BME. Can collagen I inhibit the differentiation of basal-like cells? What will happen to organoids by serial passage in collagen I?

2. The authors compared gene expression profiles between the basal-like cell populations in Collagen I and BME. However, they showed only upregulated genes without the analysis of downregulated genes (Figure 3a). What is the correlation between down-regulated genes and breast cancer invasion? A heatmap would be helpful to illustrate the of significantly different expressed genes in the two groups.

3. The authors showed that KO of K14 was not sufficient to prevent organoid invasion (Figure 4e, f). K14 KO leader cells showed a protrusive morphology and maintained nuclear localized enrichment of Yap (Figure 4g). However, this finding was very different from Reference 1, which also used cancer organoids from several mouse strains, including MMTV-PyMT organoids and the found that shRNA mediated KD either K14 or P63 disrupted organoid invasion and the formation of leader cells. It is very

difficult to understand the causes of such a big discrepancy. The current study was only based on one type of organoid. This is a very important issue need to be further investigated:

- 1) Is this also true in organoids driven by other oncogene(s)?
- 2) The organoids with KO K14 should be implanted in the mice to study its effect collective breast cancer cell invasion and cancer metastasis in vivo
- 3) Since the culture technology of human breast cancer organoids is well established and gives high success rate, can authors test whether similar phenomena can also be observed for human breast cancer-derived organoids?

4. For the monoclonal MMTV-PyMT organoid formation assay, what is the success rate of the experiment, i.e. number of organoid formed vs the total number cells seeded? Is this done by using cells from each individual organoid?

5. Authors indicated that "The presence of K8^{low}K14^{high} cells (Figure 2a, arrowheads), suggests that a subset of the luminal-derived MMTV-PyMT tumor cells gave rise to basal like cells that express myoepithelial markers. This data is not enough to support such a statement. Further details or data are required. For example, can the authors show evidence that the original organoids are K8^{high}K14^{neg}? It is important to use K8^{high}K14^{neg} for the monoclonal organoid formation study to see if they can form mixed organoids.

6. In Figure 2, the authors showed that Collagen I is neither required nor does it induce the formation of basal like cells. Based on this data, the authors concluded that "Thus, our model indicates that a basal-like cell identity and becoming a leader cell in breast cancer invasion are uncoupled processes". This conclusion is biased. This is true that not all the basal-like cells can become the leader cells, however data also indicated that only basal-like cells, but not luminal cells, can become the leader cells. Thus, a more consistent conclusion should be that the identity of basal-like is required yet only a subset of them can become the leader cells. This actually highlights a need of other ECM signaling.

7. The title: "Yap-TEAD associated transcriptional program is activated in basal-like cells in Collagen I" does not read well. ...cells treated by Collagen I? This is a key finding in this study. However, the data is quite descriptive. Further investigations regarding how the Yap-TEAD is activated in the basal cells by Collagen I might be required here.

8. Figure 6: Yap promotes local invasion in vivo and is expressed in invading strands in patients". Histopathology was used to link their finding to human breast cancer patients, which is not convincing. Organoids from human breast cancers are needed for this. KO Yap can be done in such organoids followed by implantation into the nude mice. Also, the authors are suggested to study potential effect of Yap KD on cancer metastasis in other organs.

Minor points

1. The authors analyzed the expression of K14 and K8/18 in 20 IDC samples and found most of them (19/20) contained collective invasion patterns. Their results showed only 7 of 19 samples contained K14-positive cells, whereas the collective cancer invasion of most tumors (12/19) was not caused by basal-like cells. However they concluded that "In short, our clinical results confirm findings from the MMTV-PyMT organoid model, where basal-like cells lead collective cancer invasion". This conclusion is not consistent with their data.

2. Line 117: "The underlying role of the ECM" is not a full sentence.

3. 171-176: Because a direct comparison of mRNA read-counts could not be performed between the 2 separate experiments, we compared the fold change in read counts per mRNA between basal-like and luminal cells in Collagen I vs BME (e.g. comparing the fold change in Tenascin C (Tnc) mRNA counts

between basal and luminal cell clusters in Collagen I (FC = 14.80) and BME (FC = 2.12) gives a is delta fold change (ΔFC) of 12.68 (Figure 3a).

This sentence is too long and difficult to understand. What is "...gives a is delta fold change..."? The "a" means "a different value"?

Also figure 3a has no Cldn4, but it was stated by the authors. What is Tgfb1 in figure 3a?

4. The tumor growth of Yap shRNA1+Dox feed group was bigger than the control group from 11-13 weeks, and then go down after 14 weeks (Figure 6b). How to explain this phenomenon? Does decreased Yap gene expression lead to an increased rate of tumor cell proliferation?

Reviewer #3 (Remarks to the Author):

Overall this is a scientifically strong and well-written manuscript with novel findings that help advance the breast cancer collective invasion field. It provides some mechanism around K14/Yap/collagen function during cancer invasion in an in vitro and in vivo model system. It also incorporates patient samples at the end to provide translational relevance of the findings. Imaging quality and quantitative analysis is also strong, with high scientific rigor throughout. There are several key moderate weaknesses that should be addressed to further enhance the mechanistic depth of the manuscript and to improve impact.

1- The use of a single model system throughout (organoids and in vivo) for mechanistic experiments should be expanded where the key findings are tested in other 3-D models

2- Due to the variability in collagen remodeling in Fig 4C, additional data points are warranted. While there is statistical significance, there is also wide variability, especially since this is a key finding. Additional models could help improve the impact of this finding.

3- Almost all of the mechanistic data relies on a YAP KO system. A YAP overexpression system, especially targeted to the nucleus, could test additional questions on promoting collagen remodeling due to forced YAP expression, especially nuclear YAP

4-Taz localization as well should be probed in the initial experiments to confirm localization

5- Collagen localization should be assessed in the mouse model experiments (Fig 6) to further test their proposed model in vivo

6- Fig 6H Image quality of nuclear Yap makes it difficult to visualize

7- In addition, can the patient data in Fig 6 be better well-characterized by focusing on which cell regions have nuclear YAP (invasive areas, K14 positive, etc). Better quantification of these data would improve significance.

8- Since their data show that K14 is not required for collective invasion, additional discussion around this topic and how it fits into the bigger picture (including findings by others is important). In general the discussion could be expanded to bring in other data in this field and how their works fits in.

9- Improve discussion on why more than half of the patients in 1D are K14 negative. Other mechanisms, ideas?

We thank the reviewers for their positive comments, the critical review and the suggestions to improve our manuscript entitled 'A YAP-centered mechanotransduction loop drives collective breast cancer cell invasion'. We have addressed all of the points raised by the reviewers as explained below in our point-by-point response to the reviewer's suggestions and comments.

Reviewer #1

In this manuscript, Khalil and colleagues describe the existence of a feedforward loop between ECM composition/stiffness, YAP activity and ECM remodeling in a distinct subset of doubly K8 and K14 positive cells derived from the PYMT mouse breast cancer model. This proposed loop enables these cells to acquire a "leader cell" behavior and to promote collective cell migration in the presence of collagen.

Major comments

- (1) The authors describe phenotypes that are different from similar experiments made by others, with apparently the same cell system. While this is not surprising per sé, it would be important to resolve these differences or at least highlight the possible reasons. For example, the authors use organoids derived by others and (presumably) kept for long time in culture. This might have introduced biases due to initial bottlenecks, or to selection in vitro. So, also in light of the reproducibility of the data, it would be important to repeat key experiments based on freshly-isolated PYMT cells.

To address the concern of reviewer 1, we have used fresh organoids immediately after isolation from MMTV-PyMT mammary tumors from mice. After embedding in BME or Collagen I and confocal imaging for K14 and K8, We confirmed that the basal-like cells (K14+) are present at the organoid-ECM interface in both matrices (Rebuttal Figure 1a). We also transduced the freshly isolated MMTV-PyMT organoids with lentiviral particles (Yap shRNA) and show that the transition of preexisting basal-like cells into invasive leader cells is dependent on Yap, similar to our previous results. These data using freshly isolated organoids are now included in Figure S2h and S2i. Regarding the role of K14 in invasion, we tried multiple attempts to knockout Keratin 14 in the freshly isolated MMTV-PyMT organoids, however these attempts failed due to technical difficulties. Namely, freshly isolated cells did not survive after FACS sorting, an essential step for efficient Crispr/cas9-mediated gene KO. We followed the suggestion of reviewer 1 and highlighted the possible reasons behind the differences between the results from our studies and those of other reports. We now discuss two potential reasons compensatory mechanism after K14 KO and possible loss basal-like cells in the previous report. This is now part of the discussion on page 15: *"K14-independent invasion that we observe in 3D collagen I contradicts previous reports where K14 knockdown was shown to inhibit invasion of the MMTV-PyMT organoids. Cheung et al., used lentiviral transduction of freshly isolated organoids followed by rounds of antibiotic selection to enrich for the cells that express the shRNA¹. Whether the K14 KD organoids still contain basal-like cells after the rounds of selection remains unclear. We show by p63 immunostainings that the basal-like cells are still present in the K14 KO organoids and able to drive collective invasion. We cannot exclude compensatory mechanisms (as a consequence of culturing or gene KO) that allow basal-like cells to drive invasion independent of K14, K5 and K17. However, combined with our immunohistological analysis, where collective invasion occurs without the expression of K14 in many clinical breast cancer samples, our work show that leader cells can still emerge and drive invasion without the expression of K14."*

- (2) While PYMT have been for a long time a model to study breast cancer, in reality they do not recapitulate any of the cancer phenotypes/classes that are observed in human breast cancers. So one important issue is whether these observations are specific to the PYMT model, or more general. This issue is also relevant in light of the recent ... where it was shown that the type of ECM, rather than mechanical properties, regulate K8/K14 in human patient-derived breast cancer spheroids².

To address the reviewers concern about the translational value of the MMTV-PyMT model to human breast cancer, we now include a patient derived male invasive ductal carcinoma (UDL-MBC8). Using this human organoid model we show that the organoids embedded in either BME or in collagen I have k14 positive cells located at the tumor-ECM interface. The K14 positive cells become invasive in collagen I and show high nuclear Yap levels. We further show that the transition of basal-like cells into invasive leaders is dependent on Yap-TEAD using a small molecule inhibitor K975. These data are now included in (Figure 3e, i, m) and (Figure S2j). In addition to the patient-derived organoid, we used 4T1 breast metastatic cell line as an alternative mouse model and show that overexpression of YTIP blocks invasion (Figure 3j, k). With these experiments, we confirm that the presence of basal-like cells in BME and their dependence on YAP to drive invasion in collagen I are more general phenomena and not specific to the MMTV-PyMT model.

- (3) The authors show that K14+ve basal-like cells preferentially activate YAP in response to collagen, which promotes their “leader cell” invasive action. However, K14 itself is not required for such preferential activation of YAP and “leader cell” phenotype. Based on single-cell sequencing, can the authors identify some other marker of “K14+ve” basal-like cells to ensure that the leader cells emerging in K14-KO organoids are of the same lineage, and different from the other “luminal only” cells?

We agree with the reviewers comment. In the initial manuscript we show that basal-like cells express low levels of K8. After K14 KO, the invasive strands were guided by K14-negative cells that express low levels of K8, suggesting that the leaders have still basal cell properties and are not luminal-only. To further confirm this point, we now stained K14 WT and KO organoids for p63, a major basal cell marker. In line with the low K8 expression, our data show that the K14-negative leader cells express nuclear p63, confirming that the cells are indeed of basal lineage but lack K14. These data are now included in Figure 4i, Figure S3e.

- (4) The authors discuss that their data uncover a previously unappreciated feedforward loop between ECM mechanical properties, YAP activation and further ECM remodeling. First, this is not entirely new as many observed previously similar loops in many cell types. This needs to be acknowledged.

We agree, we now mention: Yap at the center of a previously unappreciated mechanical feed forward loop in the induction of leader cell functions and collective invasion. This could explain the stochastic occurrence of invasive strands at the tumor/organoid ECM interface.

Second, this is not really demonstrated, apart from the acquisition of a leader cell morphology/position. There is no demonstration of a YAP-dependent ECM remodeling. For example, one could measure traction forces at the organic interface, and show these are higher

facing K14+ cells, and absent when YAP is inhibited. In the initial submission, we showed that the alignment of collagen fibers adjacent to basal-like cells is inhibited upon Yap knockdown (Figure 4b, c). To further confirm this point, we used another approach of assessing for ECM pulling. We embedded fluorescent beads in the collagen gel and measured their displacement towards the direction of the organoid after interfering with Yap. Compared to control, Yap knockdown in mixed MMTV-PyMT organoids caused a strong reduction in the displacement of beads towards the organoid body. On the other hand, luminal organoids that are deficient of K14 show minimal bead displacement and collagen fiber alignment. By overexpressing constitutively active Yap in luminal organoids, our data show increased bead displacement and pulling on the collagen I fibers. These experiments confirm that Yap is essential for ECM remodeling and are now presented in (Figure 4d, e; Movies 3, 4).

Moreover, there is no demonstration this specifically occurs in leader cells. One way of demonstrating this could be to engineer cells to specifically inhibit YAP when they become K14+, or to mix TAM-treated K14+ cells with K14^{low} cells without shRNA from a homogeneous organoid. Moreover, is this a cell intrinsic ability of K14+ cells observed also when they are single? Are these more contractile because of YAP?

In the initial submission, we showed that Yap is nuclear in basal-like cells and not in luminal cells, even when the luminal cells are in contact with collagen I. We now show that luminal organoids (deficient of basal-like cells) fail to pull on collagen fibers. Regarding the single cell experiments suggested by reviewer 2, we believe that these experiments will not represent ECM remodeling performed by leader cells. Leader cells maintain cell-cell junctions with the follower cells at the back and this is crucial for the polarized behavior. When individualized cells are embedded in 3D collagen I, they fail to polarize in the absence of cell-cell junctions and are predominantly rounded and move on the spot³.

Minor points:

- the authors define their 3D cultures "organoids". Are these containing other cell types besides breast cancer cells? If not, maybe it would be better call them "spheroids" as more commonly used in the cancer biology field.

The term organoids does not necessarily mean that it contains other cell types. In fact in the protocol of organoid isolation there are particular steps performed to get rid of other cell types (incl. fibroblasts, immune cells)^{1,4}. We suggest to use term tumor organoids to reflect their primary tissue origin and mostly follow this terminology, except when we use cell lines (4T1 cells), we will use the term spheroids.

- in the initial characterization, it should be made clear (perhaps, also with an IF on a normal mammary duct) whether all PYMT cells are K8+ve and a subset ALSO expresses K14, and whether this is different from the normal K8/K14 dichotomy observed in the normal duct.

We thank the reviewer for this comment. Although the K14 positive cancer cell express low levels of K8, they still maintain the K8 expression. This in contrast to the myoepithelial cells in normal ducts where K8 is absent¹. To confirm this is in our model, we now show immunostains of K8 and K14 during invasion of MMTV-PyMT cells in vivo. Single cell quantifications of K8 and K14 intensities show that both basal-like leader cells and myoepithelial cells (in adjacent normal ducts) express comparable levels of K14. However, and as expected, the basal-like cells

express higher levels of K8. We now include these data in the text on p6, line 105 and in the initial model characterization figure (Figure 1c, d).

- line 119 what are the non-invasive conditions? BME without added collagen? Please make clear in the text when first describing.

We adjusted the text as follows: To further assess the basal and luminal cell identity in non-invasive and invasive conditions, MMTV-PyMT organoids were embedded in either BME (non-invasive) or Collagen I (invasive) for 5 days and analyzed by single cell mRNA sequencing.

- bold and underlined genes in 3a should also include INHBA and CTGF that are recurrent YAP/TAZ-dependent genes in many differential expression analyses.

We agree, INHBA and CTGF are included as YAP/TAZ-dependent genes as depicted by the black squares in the new heatmap (Figure 3a).

- are K14LOW rim cells in Collagen also displaying nuclear YAP compared to the same cells in BME or not? This would be important to know whether basal-like cells are insensitive to collagen-mediated YAP activation, or just less sensitive. This could be enriched by also seeding single cells on top of BME vs collagen, to see whether such differential sensitivity relates to the multicellular environment or to intrinsic differences.

K14 low cells located at the rim of the organoid have comparable levels of nuclear Yap in BME vs Collagen (Figure 3c), suggesting that these cells indeed cannot activate Yap in response to contact with collagen I.

- what is the specificity of the TAZ antibody?? The result in S2C is unexpected, but TAZ antibodies are often non-specific and it is important to make the difference.

We have now used 2 additional new antibodies and immunostained MMTV-PyMT organoids invading 3D collagen I gels. By confocal microscopy, we found only cytosolic staining in the cells of the organoids including the basal-like cells whether located at the rim of the organoids or at the fronts of the strands (Figure S2d, e). Since 3 independent antibodies showed cytosolic distribution of TAZ in invasive strands, our data strongly suggest that Taz is not active during collective invasion.

- the effects of Yap inhibition on leader cell and organic proliferation should be pinpointed better by using more specific markers (EdU incorporation for example). In the context of collective migration proliferation does play a role in “pushing” cell strands and it would not be contradictory to show this.

In the first draft of the manuscript we show that yap knockdown or inhibition does not affect the growth of the organoid (Supplementary Figure 2l). We now confirm this using Ki67 immunostaining and show that the % of K14 positive cells that have nuclear Ki67 are similar in control and Yap-inhibited organoids. These data are now presented in Figure S2m.

- in vivo data need to be repeated with a second shRNA for YAP or with an independent way of inhibiting YAP, or with a rescue experiment. Any shRNA could have off-target effects on cell migration or any other phenotype, and the specificity of the YAP shRNA is unknown.

Despite significant effort, we were unable to achieve good YAP knockdown with additionally designed shRNA constructs. We have therefore repeated the in vivo experiments using dox-inducible expression of YTIP. Similar to Yap knockdown, YTIP expression reduced the protrusions of the tumors in vivo (Figure 6d, g), without affecting tumor growth (Figure S4b). These data suggest that the effect on invasion observed in vivo is not due to an off target effect of the Yap shRNA.

Reviewer #2 (Remarks to the Author):

In this study, using the MMTV-PyMT mammary carcinoma organoids, Khalil et al. showed that Collagen I drives a force-dependent loop, specifically in basal-like cancer cells, for collective breast cancer cell invasion. The feed-forward loop is centered around the mechanotransducer Yap and independent of K14 expression. Yap promotes a transcriptional program that enhances Collagen I alignment and tension, which further activates Yap. Active Yap is detected in invading breast cancer cells in patients and required for collective invasion in 3D collagen I and in the mammary fat pad of mice. The most noteworthy work is the finding of a new function for YAP in leader cell selection during collective cancer invasion. Although there are some new findings, which are interesting, many data are quite descriptive and some further studies are needed.

1. Their single-cell mRNA sequencing results showed that the proportion of basal-like cells was significantly lower in collagen I than in BME. Can collagen I inhibit the differentiation of basal-like cells? What will happen to organoids by serial passage in collagen I?

To address comment 1 of reviewer 2, we passaged MMTV-PyMT organoids in collagen I for 2 weeks. The passaging of organoids in 3D collagen I requires collagenase treatment (to isolate the organoids from the gel) followed by trypsinization and embedding in collagen I. In collagen I, the individualized cells and small clusters (2-10 cells) remained rounded and failed to form organoids, in contrast to BME. However, the bigger multicellular clusters developed into organoids with basal-like and luminal cells (Rebuttal Figure 1a) suggesting that differentiation of basal-like cells is still occurring during long exposure to collagen I signals. We believe that investigating the effects of collagen I on differentiation and/or proliferation of basal-like cells is interesting and requires in-depth work (incl. establishing techniques for long term propagation of organoids in 3D collagen I as well as cell cycle analysis); which we believe is out of the scope of this paper. This work focuses on the mechanism by which collagen I induce basal-like cells (already differentiated) to become invasive leaders.

The authors compared gene expression profiles between the basal-like cell populations in Collagen I and BME. However, they showed only upregulated genes without the analysis of downregulated genes (Figure 3a). What is the correlation between down-regulated genes and breast cancer invasion? A heatmap would be helpful to illustrate the of significantly different expressed genes in the two groups.

We agree with the reviewer's comment and now show the expression profiles in a heat map (Figure 3a) and include the analysis of downregulated genes (Suppl. Fig. 2b). The new analysis of downregulated genes shows that the only two genes significantly downregulated in different ECM conditions and cellular subset are *Csn3* and inhibin beta b. These two genes are downregulated in basal-like cells compared to the luminal cells in collagen I.

3. The authors showed that KO of K14 was not sufficient to prevent organoid invasion (Figure 4e, f). K14 KO leader cells showed a protrusive morphology and maintained nuclear localized enrichment of Yap (Figure 4g). However, this finding was very different from Reference 1, which also used cancer organoids from several mouse strains, including MMTV-PyMT organoids and the found that shRNA mediated KD either K14 or P63 disrupted organoid invasion and the formation of leader cells. It is very difficult to understand the causes of such a big discrepancy. The current study was only based on one type of organoid. This is a very important issue need to be further investigated:

1) Is this also true in organoids driven by other oncogene(s)?

We agree. We tested K14 KO in another model, the 4T1 metastatic cancer cell line and show that K14 KO does not inhibit neither the formation nor the function of leader cells. These data are now included in Figure 4h; Figure S3c. Combined with the patient data that show in many samples collective invasion without K14 expression, our work strongly indicate that collective invasion can take place independent of K14. In addition, we discuss in the discussion section possible reasons behind the discrepancy between our results and the results obtained from Reference 1.

2) The organoids with KO K14 should be implanted in the mice to study its effect collective breast cancer cell invasion and cancer metastasis in vivo

Although it would be interesting to check if K14 KO affects tumor invasion and metastasis in vivo, we have not performed these experiments for the following reasons:

- Scope: The main question that this work aims to answer is how collagen I induces basal-like cells to become invasive. The basal-like cells that are K14 KO still protruded and drove strand invasion in 3D collagen I and in a big subset of IDC patients (12/19). These data strongly indicate that invasion does occur without the expression of K14. Based on that, the in vivo experiments seem to us out of the scope of this paper. Such experiments would be very important to test the relevance of K14 during advanced steps of the metastatic cascade including intravasation and metastatic seeding at distant organs.
- Ethical- reduction of use of animals, based on the principles of the 3Rs (Replacement, Reduction and Refinement): Since there is no effects of K14 KO on collective invasion in vitro we prefer to reduce the number of mice. Especially, because we conclude from immunohistochemistry on patient samples that invasion does occur in the absence of K14.

- Technical: Injecting WT vs K14 KO cells in the mammary fat pad might give rise to different tumors as the starting point is different. This would result in data that are not conclusive, which also invokes the ethical consideration.

3) Since the culture technology of human breast cancer organoids is well established and gives high success rate, can authors test whether similar phenomena can also be observed for human breast cancer-derived organoids?

Whereas the technology of culturing and propagating human breast cancer organoids is indeed well established, maintaining their invasive capacity is actually not, at least not in our hands. We have tested 3 IDC organoids lines from the HUB⁽⁴⁾ and none of these invaded in collagen I. We did however establish a new human male IDC patient-derived organoids and used those to confirm the importance of YAP in basal-cell guided collective invasion of human breast cancer cells.

4. For the monoclonal MMTV-PyMT organoid formation assay, what is the success rate of the experiment, i.e. number of organoids formed vs the total number cells seeded? Is this done by using cells from each individual organoid?

We have previously included this information in the Materials and method section of the initial submission. For the generation of monoclonal cultures, MMTV-PyMT organoids were dissociated into single cells and seeded at a density of 1 cell/well in a 96-well plate containing BME. Wells containing doublets, clusters or more than one cell were disregarded. Only wells (8/96) with clear and morphologically distinct individual cells were kept and followed over a period of 5-7 days. 4/8 of the single cells grew into organoids and 3 of these monoclonal organoids were propagated and analyzed for K14 and K8 expression and invasion. We conclude from this that it is possible to select cells that form luminal organoids without any basal-like cells and that not all cells can give rise to mixed organoids (with basal-like cells).

5. Authors indicated that “The presence of K8^{low}K14^{high} cells (Figure 2a, arrowheads), suggests that a subset of the luminal-derived MMTV-PyMT tumor cells gave rise to basal like cells that express myoepithelial markers. This data is not enough to support such a statement. Further details or data are required. For example, can the authors show evidence that the original organoids are K8^{high}K14^{neg}? It is important to use K8^{high}K14^{neg} for the monoclonal organoid formation study to see if they can form mixed organoids.

We have performed an experiment that supports the claim that luminal-derived MMTV-PyMT tumor cells gave rise to basal like cells. For that, we have dissociated MMTV-PyMT organoids expressing K14 (endogenously tagged with GFP, K14 GFP, expressed in basal-like cells only) and Histone 2b mKate (expressed in luminal and basal-like cells). Dissociated single cells were sorted (FACS) as K14 negative (K14⁻) and K14 positive (K14⁺) based on GFP levels followed by seeding in BME (Rebuttal Figure 2a). Single cells were followed until day 4 where multicellular organoids are formed. Confocal imaging of the culture over time shows that both K14⁻ and K14⁺ cells give rise to organoids that eventually will contain a similar percentage of K14⁺ cells (average of 26-34 basal-like cells per organoid) (Rebuttal Figure 2c, d). These data show that luminal cells give rise to mixed organoids containing luminal and basal-like cells. Since

investigating the origin of organoids is not in the scope of this paper, we propose not to present these data in this manuscript. We changed the statement in the text to as follows: *The presence of K8^{LOW}K14^{HIGH} cells in organoids originating from single cells (Figure 2a, arrowheads), indicate that a subset of the MMTV-PyMT tumor cells may give rise to organoids with both containing both luminal and basal-like cells.*

6. In Figure 2, the authors showed that Collagen I is neither required nor does it induce the formation of basal like cells. Based on this data, the authors concluded that “Thus, our model indicates that a basal-like cell identity and becoming a leader cell in breast cancer invasion are uncoupled processes”. This conclusion is biased. This is true that not all the basal-like cells can become the leader cells, however data also indicated that only basal-like cells, but not luminal cells, can become the leader cells. Thus, a more consistent conclusion should be that the identity of basal-like is required yet only a subset of them can become the leader cells. This actually highlight a need of other ECM signaling.

We agree, we changed the statement to: Thus, our model indicates that a basal-like cell identity is required for collective invasion, where only few basal-like cells can drive multicellular movement.

We also changed the title of this result section to: “Collective invasion in Collagen I is driven by a subset of the basal-like cells located at the tumor-ECM interface”

7. The title: “Yap-TEAD associated transcriptional program is activated in basal-like cells in Collagen I” does not read well. ...cells treated by Collagen I? This is a key finding in this study. However, the data is quite descriptive. Further investigations regarding how the Yap-TEAD is activated in the basal cells by Collagen I might be required here.

We changed the title to: A Yap-TEAD associated transcriptional program is activated in basal-like cells cultured in Collagen I.

Studying the mechanism of Yap activation by collagen I is quite complex. Indeed we have tried to interfere genetically and pharmacologically with the expression of integrin expression (incl ITGb1) and found no effect on Yap activation in basal-like cells (Data not shown). We believe that several mechanosensing machineries may feed into Yap activation. Therefore, identifying the upstream players requires the targeting of many upstream candidates that seems out of scope of this manuscript. However, we now discuss how Yap may be activated in the manuscript on p. 19, line 422.

8. Figure 6: Yap promotes local invasion in vivo and is expressed in invading strands in patients”. Histopathology was used to link their finding to human breast cancer patients, which is not convince. Organoids from human breast cancers are need for this. KO Yap can be done in such organoids followed by implantation into the nude mice. Also, the arthours are suggested to study potential effect of Yap KD on cancer metastasis in other organs.

To link our finding to humans, we have now improved our histopathology experiments, by performing immunostaining and confocal microscopy on the IDC human sections, with focus on the K14 positive samples. Our analysis show that the majority (75%) of the cells with nuclear

Yap are located at the tumor-ECM interface and around 60% of those cells (nuclear Yap) are K14 positive (Figure 6i, j). We further link our finding to humans using patient derived IDC organoid (UDL-MBC8) that shows basal-guided collective invasion in collagen I and dependence on YAP for the transition of basal-like cells into leader cells. These data are now presented in Figure 3e, l, m; Figure S2k. With these new data, we do not think that an experiment using human organoids in mice will further strengthen our conclusion that Yap is important for collective invasion. Especially since we have confirmed YAP requirement for invasion in vivo using another method of YAP inhibition (YTIP) which we now include in Figure 6d, g.

We have now analyzed the effect of Yap knockdown on metastasis to the lungs and show those data in Figure S4C. Dox-inducible expression of Yap shRNA1, shRNA2 or YTIP do not affect the number and size of metastasis. We now include these data in Figure S4c.

Minor points

1. The authors analyzed the expression of K14 and K8/18 in 20 IDC samples and found most of them (19/20) contained collective invasion patterns. Their results showed only 7 of 19 samples contained K14-positive cells, whereas the collective cancer invasion of most tumors (12/19) was not caused by basal-like cells. However they concluded that “In short, our clinical results confirm findings from the MMTV-PyMT organoid model, where basal-like cells lead collective cancer invasion”. This conclusion is not consistent with their data.

We changed the text to: In short, the MMTV-PyMT organoid model recapitulates basal-lead collective invasion in a subset of IDC patients.

2. Line 117: “The underlying role of the ECM” is not a full sentence.

We adjusted this typo.

3.171-176: Because a direct comparison of mRNA read-counts could not be performed between the 2 separate experiments, we compared the fold change in read counts per mRNA between basal-like and luminal cells in Collagen I vs BME (e.g. comparing the fold change in Tenascin C (Tnc) mRNA counts between basal and luminal cell clusters in Collagen I (FC = 14.80) and BME (FC = 2.12) gives a is delta fold change (ΔFC) of 12.68 (Figure 3a).

This sentence is too long and difficult to understand. What is “...gives a is delta fold change...”? The “a” means “a different value”?

Also figure 3a has no Cldn4, but it was stated by the authors. What is Tgfb1 in figure 3a?

We removed this statement and updated the analysis and description of our single cell sequencing data on page 9 and Figure 3a and S2b.

We apologize for the typo and removed Cld4. Tgfb1: TGF beta induced

4. The tumor growth of Yap shRNA1+Dox feed group was bigger than the control group from 11-13 weeks, and then go down after 14 weeks (Figure 6b). How to explain this phenomenon? Does decreased Yap gene expression lead to an increased rate of tumor cell proliferation?

We agree with the reviewers comment, as of week 11 the tumors harboring dox-inducible YAP-

shRNA1 are bigger in dox treated mice. To test whether we see similar phenomenon in the tumors harboring dox-inducible YAP shRNA 2 and YTIP, we analyzed the growth of the tumors . Unlike Yap shRNA1, YAP shRNA2 and YTIP organoids do not show any trend of increased growth upon Dox treatment (Rebuttal Figure 3). Moreover, our in vitro data show no effect on organoid growth upon Yap knockdown and inhibition. We are thus not able to explain this increased growth phenomenon observed in vivo. We now mention this phenomenon in the result section: *“Dox-treatment did not affect tumor growth as monitored by the increase in the tumor volume over time except for the slight increase in tumor volume observed as of week 11 in the dox-treated mice (Figure 6b).* In addition, we included the growth curve of the tumors harboring YTIP in Figure S4b.

Reviewer 3 (Remarks to the Author):

Overall this is a scientifically strong and well-written manuscript with novel findings that help advance the breast cancer collective invasion field. It provides some mechanism around K14/Yap/collagen function during cancer invasion in an in vitro and in vivo model system. It also incorporates patient samples at the end to provide translational relevance of the findings. Imaging quality and quantitative analysis is also strong, with high scientific rigor throughout. There are several key moderate weaknesses that should be addressed to further enhance the mechanistic depth of the manuscript and to improve impact.

1- The use of a single model system throughout (organoids and in vivo) for mechanistic experiments should be expanded where the key findings are tested in other 3-D models.

We agree with reviewer 3, we now included two additional models to validate our major findings. We used the patient-derived male IDC organoids and 4T1 breast cancer model to confirm the importance of YAP in collective cancer cell invasion. These data are now presented in Figure 3j-m and Figure S2k)

2- Due to the variability in collagen remodeling in Fig 4C, additional data points are warranted. While there is statistical significance, there is also wide variability, especially since this is a key finding. Additional models could help improve the impact of this finding.

We agree. We now used another approach of assessing for ECM pulling. We embedded fluorescent beads in the collagen gel and measured their displacement towards the direction of the organoid after interfering with Yap. Yap knockdown in mixed MMTV-PyMT organoids caused a strong reduction in the displacement of beads towards the organoid body. Conversely, overexpressing constitutively active Yap in luminal organoids (Basal-cell deficient) induced pulling of the collagen I fibers beads. These data demonstrate that Yap is essential for ECM remodeling and are now presented in Figure 4d, e; Movie 3, 4).

3- Almost all the mechanistic data rely on a YAP KO system. A YAP overexpression system, especially targeted to the nucleus, could test additional questions on promoting collagen remodeling due to forced YAP expression, especially nuclear YAP

We agree, this has been addressed in our reply to comment 2

4-Taz localization as well should be probed in the initial experiments to confirm localization

We have immunostained for TAZ using 3 different antibodies and show that it is located in the cytosol (Suppl. Figure 2d, e).

5- Collagen localization should be assessed in the mouse model experiments (Fig 6) to further test their proposed model in vivo

.

The primary tumors in our mouse models are isolated at the end point, which is after ~14 weeks of tumor growth and invasion into the mammary tissue. This is very different from the organoids embedded in collagen I for few days. However, we still performed this experiment, as we believed that the reviewer's suggestion is valid/important. For that, we have imaged collagen I by trichrome staining in the primary tumors from control and DOX-treated mice (Yap shRNA). We found that in both control and Dox-treated mice the predominant orientation of the collagen fibers was tangential relative to the tumor border (Rebuttal Figure 1c, yellow arrowheads). Occasionally we found in the control mice (-DOX) some regions with collagen I fibers aligned parallel to the invasive strands (Rebuttal figure 1c, green rectangle). Since the alignment events were rare due to the excessive growth of the tumor, we decided not to proceed with further collagen I analysis. We believe that this is due to the long time points of the in vivo experiments (up to 14 weeks) where tumors grow much and replace most of the stroma. New in vivo experiments with shorter time points would be needed to assess for the collagen alignment induced by the invasive strands.

6- Fig 6H Image quality of nuclear Yap makes it difficult to visualize

We replaced the image with one with a better quality, it is now presented in Figure S4e.

7- In addition, can the patient data in Fig 6 be better well-characterized by focusing on which cell regions have nuclear YAP (invasive areas, K14 positive, etc). Better quantification of these data would improve significance.

We agree with the reviewer's comment. We now immunostained the patient sections for Yap, K14 and K8. We focused on the K14 positive samples and quantified which cells have nuclear yap relative to their position and K14 positivity. We found that 75 % of the cells with nuclear Yap are actually located at the borders of the multicellular groups and 60% are K14 positive. These data are now included in Figure 6i, j. The data confirm our in vitro assays that show increased Yap activity in the basal-like cells that are in contact with the ECM and lead collective invasion.

8- Since their data show that K14 is not required for collective invasion, additional discussion around this topic and how it fits into the bigger picture (including findings by others is important). In general the discussion could be expanded to bring in other data in this field and how their works fits in.

We have included a discussion (p. 17 line 379) about the role of K14 in invasion.

9- Improve discussion on why more than half of the patients in 1D are K14 negative. Other mechanisms, ideas?

We now refer to other studies that show K14-negative tumors (p. 17 line 387) and discuss that several of the genes that are upregulated in basal-like cells (incl. Vim, Tpm2, Tnnt2, Pls3) regulate actin protrusions and may contribute to cytoskeletal changes that accommodate Collagen I pulling and alignment (p.18 line 406).

Rebuttal Figure 1

Rebuttal Figure 2

Rebuttal Figure 3

Tumor growth in vivo (week 11)

References

1. Cheung, K. J., Gabrielson, E., Werb, Z. & Ewald, A. J. Collective invasion in breast cancer requires a conserved basal epithelial program. *Cell* **155**, 1639–1651 (2013).
2. Munne, P. M. *et al.* Compressive stress-mediated p38 activation required for ER α + phenotype in breast cancer. *Nat Commun* **12**, 6967 (2021).
3. Khalil, A. A. *et al.* Collective invasion induced by an autocrine purinergic loop through connexin-43 hemichannels. *Journal of Cell Biology* **219**, (2020).
4. Sachs, N. *et al.* A Living Biobank of Breast Cancer Organoids Captures Disease Heterogeneity. *Cell* **172**, 373-386.e10 (2018).

REVIEWER COMMENTS

Reviewer #1 (Remarks to the Author):

The authors performed most of the requested experiments and their results reinforce the previous claims. An issue remains on the lack of stronger YAP loss-of-function evidence, for which I would recommend carefully describing the results and discussing the implications in the final version of the manuscript.

Sirio Dupont

Reviewer #2 (Remarks to the Author):

The authors addressed some of my concerns, however left a few major issues either not answered or unsatisfactorily addressed, which are listed below.

Q2. Although the authors added down-regulated genes in Suppl. Fig. 2b, it remains difficult to understand the overall landscape in gene expression after collagen I and BME culture of basal cells. The authors should have data to show all up- and down-regulated genes and related signaling pathways.

Q3-3). The authors indicated that “We did however establish a new human male IDC patient-derived organoids and used those to confirm the importance of YAP in basal-cell guided collective invasion of human breast cancer cells”. But, where are the data?

Q5. We want to see the evidence that K8high K14neg cells (a subset of the luminal-derived MMTV-PyMT tumor cells) could give rise to basal-like cells (K8low K14pos) or mixed organoids. But, in the rebuttal figure 2, the author employed H2b pos and K14 neg markers to identify luminal cells; thus, the entire set of rebuttal experiments lacks information on K8 expression. Because H2b and K8 are different markers, they are not equivalent for cell labeling, So, I recommend continuing to use K8 instead of H2b to complete the rebuttal experiment.

Also, to eliminate interference between cells, it is crucial to utilize K8^{high}K14^{neg} single cell in the monoclonal organoid formation study to assess their potential for generating mixed organoids.

Q7. While it is difficult to investigate how type I collagen activates Yap-TEAD in basal cells, this is the main finding of this study and the authors should investigate this mechanism.

Reviewer #3 (Remarks to the Author):

The reviewers have addressed the previous concerns and propose that this is suitable for publication.

We thank the reviewers for their constructive comments and suggestions to improve our manuscript entitled ‘A YAP-centered mechanotransduction loop drives collective breast cancer cell invasion’. We have addressed all of the points raised by the reviewers as explained below in our point-by-point response.

Reviewer #1

The authors performed most of the requested experiments and their results reinforce the previous claims. An issue remains on the lack of stronger YAP loss-of-function evidence, for which I would recommend carefully describing the results and discussing the implications in the final version of the manuscript.

We thank reviewer 1 for his constructive comments. We have now included in the result section on page 15 (line 337 and line 343) the % of inhibition of invasion observed in vivo after interference with Yap. Furthermore, we discuss on page 18 line 405 the possible reasons behind the lack of stronger effects after Yap inhibition and implications of targeting Yap complex tissues.

Reviewer #2 (Remarks to the Author):

The authors addressed some of my concerns, however left a few major issues either not answered or unsatisfactorily addressed, which are listed below.

Q2. Although the authors added down-regulated genes in Suppl. Fig. 2b, it remains difficult to understand the overall landscape in gene expression after collagen I and BME culture of basal cells. The authors should have data to show all up- and down-regulated genes and related signaling pathways.

We thank reviewer 2 for his persistence in pushing us to get the most information out of the single cell sequencing data. With the help of experienced researchers, we have therefore reanalyzed our single cell sequencing data using newer versions of the statistical analysis methodology. This improved data normalization and we now show UMAP plots of the clustering analysis in BME vs Collagen and volcano plot showing the downregulated and upregulated genes in basal-like cells in Collagen I compared to BME. Furthermore, we performed transcription factor motif analysis and pathway analysis. These data are now included in Figure 2f, g and Figure 3a, b, c. We also include now all the regulated genes as excel sheet (Supplementary table).

Q3-3). The authors indicated that “We did however establish a new human male IDC patient-derived organoids and used those to confirm the importance of YAP in basal-cell guided collective invasion of human breast cancer cells”. But, where are the data?

The data from human male IDC patient-derived organoids are shown in Figure 3g, m, n and Figure S2k.

Q5. We want to see the evidence that K8^{high} K14^{neg} cells (a subset of the luminal-derived MMTV-PyMT tumor cells) could give rise to basal-like cells (K8^{low} K14^{pos}) or mixed organoids. But, in the rebuttal figure 2, the author employed H2b pos and K14 neg markers to identify luminal cells; thus, the entire set of rebuttal experiments lacks information on K8 expression. Because H2b and K8 are different markers, they are not equivalent for cell labeling, So, I recommend continuing to use K8 instead of H2b to complete the rebuttal experiment. Also, to eliminate interference between cells, it is crucial to utilize K8^{high}K14^{neg} single cell in the monoclonal organoid formation study to assess their potential for generating mixed organoids.

To address the comment of reviewer 2, we have dissociated MMTV-PyMT organoids expressing endogenously tagged K14 (GFP) into single cells (Rebuttal figure 1a). Single cells were then sorted (FACS) based on GFP expression as K14-negative (K14-ve) or K14-positive (K14+ve). Then we embedded the cells in BME for 0, 4 and 8 days followed by fixation and immunostaining for Keratin 8 (K8). We show that all the organoids generated from K14 -ve single cells contain K14+ve cells after 4 and 8 days of culture (Rebuttal figure 1b, c). Furthermore, all the K14-ve cells are actually K8 positive at day 0 (Rebuttal Figure 1b, c), indicating that the K14-ve cells are K8 positive and could give rise to organoids comprising of luminal and basal-like cells (mixed organoids). We feel that we have addressed the comment with this experiment and thank the reviewer for the incentive to support our conclusions in a more direct manner.

Q7. While it is difficult to investigate how type I collagen activates Yap-TEAD in basal cells, this is the main finding of this study and the authors should investigate this mechanism.

We agree with the reviewer that uncovering the mechanism would be highly interesting, however, we also find that, given the contradictions and complexity in the field currently, deciphering exactly how collagen I activates Yap will be a daunting task that is outside the scope of this current study. We discuss potential mechanisms in the discussion on page 19 line 433.

Rebuttal figure 1

a

b

c

REVIEWERS' COMMENTS

Reviewer #2 (Remarks to the Author):

I am fine with authors' response to my comments.

To the reviewers of our manuscript entitled "A YAP-centered mechanotransduction loop drives collective breast cancer cell invasion",

Explanation and description of revisions after acceptance

As discussed with and requested by the editor, we have revised our manuscript after it was accepted. The reason is that we detected an unfortunate mix-up in the early stages of the derivation of the MBC8 patient-derived organoid line. We discovered this discrepancy during routine genomics analyses meant to ensure organoid purity. Contrary to the initial characterization, we have to report that the MBC8 organoids used in our study are not derived from male invasive ductal breast carcinoma. Based on comparative STR analyses on different passages and model systems in our laboratories, we were able to trace back the origin of the MBC8 organoid to a female patient-derived head and neck squamous cell carcinoma (HNSCC).

We extensively explored various human breast cancer patient-derived organoids, including those generated in the Clevers lab¹. However, none exhibited Keratin 14 expression or demonstrated invasion in 3D collagen I, except for MBC08. It stands out as the sole human "breast" cancer organoid expressing Keratin 14 and exhibiting invasiveness in 3D collagen I. Importantly, while only few studies in the literature employed human organoids, they typically utilized fresh tumor pieces to demonstrate the presence of basal cells or invasion in 3D collagen I². However, most if not all of the existing literature using breast cancer organoid cultures in invasion studies originate from mice. This, underscores the well-documented challenges in maintaining fidelity between human breast cancer tissue and organoid culture³. The HNSCC organoid line was used in Figure 3g, m, n, where we aimed to confirm the role of Yap in leader cell formation using human cells in vitro. This addition to the study was made in response to your comments during the review process, specifically addressing the generalizability of our observations beyond the PYMT model and the possibility of observing similar phenomena in human breast cancer-derived organoids.

Instead of the use of human invasive breast cancer organoids, that we will not be able to provide, we do feel that the experiments conducted with the HNSCC organoid line address a significant aspect of the question: the importance of YAP-dependent invasive transformation of basal cells in human cancer. Supported by experiments in two other mouse breast cancer models (MMTV-PyMT and 4T1) and validation in human tissue sections from breast cancer patients, our findings validate as much as possible the fundamental mechanistic insights of our study in humans. The literature on collective invasion in both invasive breast cancer and head and neck tumors, orchestrated by basal cells^{4,5}, further supports this assertion.

To rectify this situation, we have revised the results section to provide a clear and concise explanation of the choice of the human tumor organoid. Emphasizing the lack of availability of human breast cancer organoids expressing Keratin 14, we explain the scientific significance of utilizing H&N cancer organoids as a relevant alternative to study basal-guided collective invasion of human cancer cells in collagen I.

Despite the mix-up and lack of an exact replacement of the model, we remain confident that the fundamental conclusions drawn from our study are robust and generally applicable. Your insights during the review process were invaluable, and we appreciate your extra time in dealing with this unforeseen circumstance.

Best regards,

Johan de Rooij and Antoine Khalil

REFERENCES

1. Sachs, N. et al. A Living Biobank of Breast Cancer Organoids Captures Disease Heterogeneity. *Cell* 172, 373-386.e310 (2018).
2. Cheung, K.J., Gabrielson, E., Werb, Z. & Ewald, A.J. Collective invasion in breast cancer requires a conserved basal epithelial program. *Cell* 155, 1639-1651 (2013).
3. Goldhammer, N., Kim, J., Timmermans-Wielenga, V. & Petersen, O.W. Characterization of organoid cultured human breast cancer. *Breast Cancer Res* 21, 141 (2019).
4. Sterz, C.M. et al. A basal-cell-like compartment in head and neck squamous cell carcinomas represents the invasive front of the tumor and is expressing MMP-9. *Oral Oncol* 46, 116-122 (2010).
5. Okuyama, K. et al. Anaplastic transition within the cancer microenvironment in early-stage oral tongue squamous cell carcinoma is associated with local recurrence. *Int J Oncol* 53, 1713-1720 (2018).

Reviewers' comments:

Reviewer #1 (Remarks to the Author):

After acceptance of the manuscript, the authors realized that they had misidentified the sole (male) breast cancer organoid that they had been using to support the existence of their proposed model in humans. This is actually a female organoid originating from a (female) head and neck cancer patient.

It is of merit that the authors disclosed this information. Of course, they should have checked the identity of the material BEFORE the very first submission, but errors can happen and it is important to acknowledge them. I guess not many would have done so.

However, this now raises an issue on the significance of the data. The data's title, abstract, and main focus remain on breast cancer. But the sole validation in humans is not.

In addition, it remains unclear whether the data on PYMT-derived cells derive from using a single isolate (as one would deduce from the fact that they obtained these cells from a colleague), or from independent isolates from independent mice. This confusion is also generated by the fact that the authors refer to "organoids", even in the case of the human HNCC, but in reality, they are using a single isolate.

In other words, the whole manuscript may be grounded on two single primary isolates, one from a PYMT mouse, and another from a human HNCC patient.

If this is the case, this is a problem because (1) even in the case of PYMT-derived cell lines, it is well known that different isolates can display different phenotypes and (2) the generality of the findings is weak, taking into consideration my old concern on the poor relevance of the PYMT system to model breast cancer.

Ideally, the authors should have used freshly-isolated tumor pieces, as in Munne Nature Communications 2021. This would have been appropriate also to explore the possibility, based on Munne et al, that the type of ECM rather than mechanical properties regulate the phenotype.

I also realize that the time-frame to perform such an additional revision might be complicated. So, one possible solution is to *very clearly* state, starting from the title and abstract, that the manuscript is limited to the mouse PYMT model of breast cancer, so that it is clear to any reader that the authors did *not* demonstrate their proposed model on other mouse breast cancer models and on human breast cancer cells. This is important also because this manuscript could be used, by a superficial reader, to question the validity of Munne et al., which is clearly not the case based on the present data. I leave to the editor to judge on whether this would be suitable for a publication in Nat Comms.

Reviewer #2 (Remarks to the Author):

The change of the identity of the MBC8 cells, from originally derived from a male invasive ductal breast carcinoma to a female patient-derived head and neck squamous cell carcinoma (HNSCC), is beyond the acceptable level for the following reasons.

1. During the reviewing process, reviewer 1 challenged whether their model from PYMT is only specific to mouse, and whether the regulation of K8/K14 they reported also occurs in human patient-derived breast cancer spheroids. Reviewer 2 requested the authors to provide human data to support their findings. By including the human patient-derived breast cancer organoid (PDO) UDL-MBC8, they effectively addressed concerns of both reviewers. However, this might not be the case if the reviewers knew the MBC8 was from HNSCC but not from a breast cancer.

2. The authors indicated that “We extensively explored various human breast cancer patient-derived organoids, including those generated in the Clevers lab 1. However, none exhibited Keratin 14 expression or demonstrated invasion in 3D collagen I”. This may indicate what they found in the mouse is not true in human, casting a challenge to their findings and conclusions. It is more likely that their finding is model specific, which might diminish the interesting of the readers.

A minor point

In the Materials and Methods, the authors indicated that MBC8 was established from an invasive ductal carcinoma that was resected from an 88-year old male patient. The tumor was poorly differentiated, ER/PR positive and HER2 negative. This description is quite clear in the last revision, which convince the reviewers that they obtained a correct cell line. The authors did not explain how this big mistake was made.

Reviewers' comments:

Reviewer #1 (Remarks to the Author):

After acceptance of the manuscript, the authors realized that they had misidentified the sole (male) breast cancer organoid that they had been using to support the existence of their proposed model in humans. This is actually a female organoid originating from a (female) head and neck cancer patient.

It is of merit that the authors disclosed this information. Of course, they should have checked the identity of the material BEFORE the very first submission, but errors can happen and it is important to acknowledge them. I guess not many would have done so.

However, this now raises an issue on the significance of the data. The data's title, abstract, and main focus remain on breast cancer. But the sole validation in humans is not.

In addition, it remains unclear whether the data on PYMT-derived cells derive from using a single isolate (as one would deduce from the fact that they obtained these cells from a colleague), or from independent isolates from independent mice. This confusion is also generated by the fact that the authors refer to "organoids", even in the case of the human HNCC, but in reality, they are using a single isolate.

In other words, the whole manuscript may be grounded on two single primary isolates, one from a PYMT mouse, and another from a human HNCC patient.

If this is the case, this is a problem because (1) even in the case of PYMT-derived cell lines, it is well known that different isolates can display different phenotypes and (2) the generality of the findings is weak, taking into consideration my old concern on the poor relevance of the PYMT system to model breast cancer.

Ideally, the authors should have used freshly-isolated tumor pieces, as in Munne Nature Communications 2021. This would have been appropriate also to explore the possibility, based on Munne et al, that the type of ECM rather than mechanical properties regulate the phenotype.

I also realize that the time-frame to perform such an additional revision might be complicated. So, one possible solution is to **very clearly** state, starting from the title and abstract, that the manuscript is limited to the mouse PYMT model of breast cancer, so that it is clear to any reader that the authors did **not** demonstrate their proposed model on other mouse breast cancer models and on human breast cancer cells. This is important also because this manuscript could be used, by a superficial reader, to question the validity of Munne et al., which is clearly not the case based on the present data. I leave to the editor to judge on whether this would be suitable for a publication in Nat Comms.

Response to Reviewer 1

We appreciate Reviewer 1 for the comments and fully understand the concerns raised regarding the relevance of our findings to human breast cancer without confirmation in a human model. It encouraged us to intensify our search for invasive human breast cancer organoids that better recapitulate invasive human breast cancers than the current models.

Fortunately, we have been able to use a new breast cancer patient-derived organoid model (PDXO 1915), which harbors basal-like characteristics and invades in 3D collagen I. Invasion of the PDXO 1915 organoids occurred as cohesive multicellular strands guided by K14 positive leader cells, similar to the collective invasion patterns observed in several human patient samples and to the MMTV-PyMT *in vitro* model.

In alignment with the previously accepted experiments, we conducted a number of assays with the human breast cancer organoid model PDXO 1915. Our results confirm that collective invasion is indeed dependent on Yap activation in this human breast cancer model. We have included these additional findings (Figure 3g, m, n; Figure S2k) to replace the data from the human HNSCC organoids. We trust that these new experiments using human breast cancer organoids address Reviewer 1's concerns and strengthen the confidence in the relevance of our findings. We have not included the HNSCC data in this manuscript in order to maintain flow and conciseness. We do feel the extension of our findings to a different human cancer strengthens the notion of broad relevance of this YAP-centered mechanism and aim to publish these findings elsewhere. If the editor and reviewers deem it necessary, we can include the data on the human HNSCC organoid as supplemental data.

We also would like to clarify that in this work, we examined the role of Yap in driving collective invasion within 3D collagen I matrices using two distinct tumor isolates sourced from separate MMTV-PyMT mice, thus ensuring the inclusion of two independent tumor isolates. One isolate was obtained from the mouse as a low-passage culture, while the second was freshly provided immediately postmortem (Figure S2g, h). In addition to the MMTV-PyMT model, we confirmed our findings using a metastatic breast cancer cell line model, 4T1 (Figure 3l, S2j, Figure 4h, S3c). Furthermore, we conducted supplementary experiments demonstrating Yap activity in a mouse APC-mutant breast cancer model¹, we are including these data in Rebuttal Figure 1. Thus, the conclusions drawn in this manuscript are not solely reliant on findings from a single mouse model, but rather are supported by evidence from multiple mouse and human primary breast tumor experimental systems.

Rebuttal Figure 1. Confocal imaging of Yap, K14 and K8 in APC mutant mouse breast cancer organoids embedded Collagen I (3 days). Arrowheads, leading cells with nuclear Yap.

In response to Reviewer 1 suggestion of using fresh tumor pieces and referring to phenotypical regulation: K14 induction in fresh mouse and human breast tumor pieces growing in 3D ECM was also found in earlier work by Ewald's group². In our view, this observation of luminal to basal reprogramming relates to a different tumor-development process than the induction of invasion in basal-like cells, which we have studied in this work. Therefore, experiments along that line as suggested by the reviewer are outside of the scope of this study.

To highlight this important point of Reviewer 1 and acknowledge previous important work on luminal to basal reprogramming, we have incorporated the following statement into the discussion on page 16 line 364: "Therefore, we focused on elucidating the mechanisms promoting the transition of pre-existing basal-like cells into invasive leader cells, rather than mechanisms driving luminal to basal reprogramming^{2,3}".

Response to Reviewer 2

Reviewer #2 (Remarks to the Author):

The change of the identity of the MBC8 cells, from originally derived from a male invasive ductal breast carcinoma to a female patient-derived head and neck squamous cell carcinoma (HNSCC), is beyond the acceptable level for the following reasons.

1. During the reviewing process, reviewer 1 challenged whether their model from PYMT is only specific to mouse, and whether the regulation of K8/K14 they reported also occurs in human patient-derived breast cancer spheroids. Reviewer 2 requested the authors to provide human data to support their findings. By including the human patient-derived breast cancer organoid (PDO) UDL-MBC8, they effectively addressed concerns of both reviewers. However, this might not be the case if the reviewers knew the MBC8 was from HNSCC but not from a breast cancer.

We agree with Reviewer 2 and appreciate the fact that we were encouraged to keep searching for invasive human breast cancer organoids. As a result, we have extended our investigation by incorporating experiments using human patient-derived breast cancer organoids (PDXO 1915). These organoids exhibit invasive behavior in 3D collagen I, forming cohesive multicellular strands guided by K14 positive leader cells, reminiscent of collective invasion patterns observed in several human patient samples and the MMTV-PyMT in vitro model. Consistent with our previously accepted experiments, we conducted several assays using these human breast cancer organoids. Our results reaffirm the dependency of collective invasion on Yap activation in this human breast cancer model. We have integrated these additional findings (Figure 3g, m, n; Figure S2k). We are confident that these new experiments using human breast cancer organoids effectively address Reviewer's 2 concern by raising confidence in the relevance of our findings.

2. The authors indicated that "We extensively explored various human breast cancer patient-derived organoids, including those generated in the Clevers lab 1. However, none exhibited Keratin 14 expression or demonstrated invasion in 3D collagen I". This may indicate what they found in the mouse is not true in human, casting a challenge to their findings and conclusions. It is more likely that their finding is model specific, which might diminish the interesting of the readers.

We appreciate the reviewer's comment and have indeed contemplated this possibility extensively. However, we emphasize that our investigation into K14-driven collective invasion was initially grounded on observations from human patients, both from previous research and our own findings. K14 positive breast cancer cells have been shown to drive invasion in patient samples and multiple breast cancer models². Our manuscript now includes experiments using a new breast cancer patient-derived organoid line (PDXO 1915). These new data (Figure 3g, m, n; Figure S2k) demonstrate that human organoids indeed recapitulate the invasion patterns observed in breast cancer patients, including the K14 driven collective invasion in 3D collagen I and similar to what we observed using the mouse breast cancer organoids.

By including these human organoid lines in our study, we have reinforced the translational relevance of our findings and their applicability to human tumor biology. We believe this clarification addresses the concern raised by the reviewer and strengthens the significance of our conclusions.

A minor point

In the Materials and Methods, the authors indicated that MBC8 was established from an invasive ductal carcinoma that was resected from an 88-year old male patient. The tumor was poorly differentiated, ER/PR positive and HER2 negative. This description is quite clear in the last revision, which convince the reviewers that they obtained a correct cell line. The authors did not explain how this big mistake was made.

The mix up happened early on when the male IDC organoids were being expanded. The exact cause of error is difficult to pinpoint, but most likely stems from a labeling mistake during the simultaneous culturing of both male IDC and HNSCC samples.

References

1. Gaspar, C. *et al.* A targeted constitutive mutation in the Apc tumor suppressor gene underlies mammary but not intestinal tumorigenesis. *PLoS Genet* **5**, (2009).
2. Cheung, K. J., Gabrielson, E., Werb, Z. & Ewald, A. J. Collective invasion in breast cancer requires a conserved basal epithelial program. *Cell* **155**, 1639–1651 (2013).
3. Munne, P. M. *et al.* Compressive stress-mediated p38 activation required for ER α + phenotype in breast cancer. *Nat Commun* **12**, 6967 (2021).

REVIEWERS' COMMENTS

Reviewer #2 (Remarks to the Author):

I have indicated in the first round of reviewing that the study is quite descriptive and request more supporting data, especially data from human breast cancer. At the second round of submission, when it ends out that a wrong cell line, MBC8 from HNSCC but not breast cancer, was used at the second of submission, it is unacceptable. I am not sure if the replacement of a breast cancer cell line can really compensate for this defect. I tried to check the information about PDXO 1915 provided the the authors, but could not figure out what this line is. The only description is “The triple negative female breast cancer patient-derived organoids PDXO 1915 were obtained from the Park Lab (McGill University, Canada)²⁵.” I also checked the reference 25, but found this paper did not mention anything about this line. As there are many subtypes of breast cancer, it is hard to believe the authors could address the questions raised by reviewers using only one line without a clear background information. The authors indicated that “By including these human organoid lines in our study...”. Guess PDXO 1915 should be “one line” but not “lines”. More lines are needed to provide stronger evidence.

Reply to Reviewer 2.

I have indicated in the first round of reviewing that the study is quite descriptive and request more supporting data, especially data from human breast cancer. At the second round of submission, when it ends out that a wrong cell line, MBC8 from HNSCC but not breast cancer, was used at the second of submission, it is unacceptable. I am not sure if the replacement of a breast cancer cell line can really compensate for this defect. I tried to check the information about PDXO 1915 provided the the authors, but could not figure out what this line is. The only description is “The triple negative female breast cancer patient-derived organoids PDXO 1915 were obtained from the Park Lab (McGill University, Canada)²⁵.” I also checked the reference 25, but found this paper did not mention anything about this line. As there are many subtypes of breast cancer, it is hard to believe the authors could address the questions raised by reviewers using only one line without a clear background information. The authors indicated that “By including these human organoid lines in our study...”. Guess PDXO 1915 should be “one line” but not “lines”. More lines are needed to provide stronger evidence.

We thank reviewer 2 for his comments and understand his/her critical comments. We have added more information about the patient-derived breast cancer organoid in the Methods section (Page 20, line 455) including information about the histological and molecular subtype. We also mention the exact name of the organoid (GCRC1915Tc) as used in the article that established this organoid line (Reference 25, Savage et al., 2020). Regarding the need of more human breast organoid lines and the editors request to downtone our relevance claims, we have included a few sentences in the discussion section (Page 18, line 425) explaining that, unlike human breast cancer in vivo, very few human breast cancer organoids contain basal-like cells, which limited our options to validate wide relevance. To further substantiate the relevance of our findings, additional basal-like cell-containing human breast cancer organoid lines need to be introduced into the field.